



# Methods and uncertainty-estimations of 3D structural modelling in crystalline rocks: A case study

Raphael Schneeberger[1], Miguel de La Varga[2], Daniel Egli[1], Alfons Berger[1], Florian Kober[3], Florian Wellmann[2], & Marco Herwegh[1]

[1]Institute for Geological Sciences, University of Bern, Baltzerstrasse 1+3, CH-3012 Bern
[2]Graduate School AICES, RWTH Aachen University, Schinkelstrasse 2, D-52062 Aachen
[3]Nagra, Hardstrasse 73, CH-5430 Wettingen

*Correspondence to*: Raphael Schneeberger (raphael.schneeberger@geo.unibe.ch)







**Abstract.** Exhumed basement rocks are often dissected by faults, the latter controlling physical parameters such as rock strength, porosity, or permeability. Knowledge on the three dimensional (3D) geometry of the fault pattern and its continuation with depth is therefore of paramount importance for projects of applied geology (e.g. tunnelling, nuclear waste disposals) in crystalline bedrock. The central Aar massif (Central Switzerland) serves as study area, where we investigate the

3D geometry of the Alpine fault pattern by means of both surface (fieldwork and remote sensing) and underground ground (mapping of the Grimsel Test Site) information. The fault zone pattern consists of planar steep major faults (kilometre-scale) being interconnected with secondary relay faults (hectometre-scale). Starting with surface data, we present a workflow for structural 3D modelling of the primary faults based on a comparison of three extrapolation approaches based on: a) field data, b) Delaunay triangulation and c) a best fitting moment of inertia analysis. The quality of these surface-data-based-3D

models is then tested with respect to the fit of the predictions with the underground appearance of faults. All three extrapolation approaches result in < 6% distance misfit when compared with underground rock laboratory mapping. Subsequently, we performed a statistically interpolation based on Bayesian inference in order to validate and further constrain the uncertainty of the extrapolation approaches. This comparison indicates that fieldwork at the surface is key for accurately constraining the geometry of the fault pattern enabling a proper extrapolation of major faults towards depth.

Considerable uncertainties, however, persist with respect to smaller-sized secondary structures because of their limited spatial extensions and unknown reoccurrence intervals.

## 1. Introduction

Geological information is inherently three dimensional (3D) in space, but often represented in 2D (Jones et al., 2009). With increasing available computer power, 3D modelling or geometrical visualizations became widespread, as they can be

performed on a desktop computer (e.g. Bistacchi et al., 2008; Caumon et al., 2009; Hassen et al., 2016; Sausse et al., 2010; Stephens et al., 2015). 3D models widely serve as basis for subsequent investigations such as stress modelling or fluid flow modelling (e.g. Hassen et al., 2016; Stephens et al., 2015). Two different approaches are mainly used for modelling 3D geological problems: explicit modelling or implicit modelling. An implicit model is built by an interpolation function defining a solid throughout space. In contrast, an explicit model uses a patchwork of 3D meshes formed by interconnected

triangles (Cowan et al., 2003). Implicit modelling allows for faster re-calculation of the model and thus faster model updating with new data (Lindsay et al., 2012). Explicit modelling, however, allows for easier visualization. Structural modelling can further be subdivided into stochastic and deterministic models. Deterministic approaches try to represent the actual occurrence of geological features, analogous to drawing a map, producing a single output (e.g. Stephens et al., 2015), whereas as in stochastic approaches parameters are defined by a probabilistic density function. Therefore, the thereout

derived mathematical instances are also probabilistic distributions (e.g. González-Garcia and Jessell, 2016; Jørgensen et al., 2015; Koike et al., 2015).

When modelling a certain volume of Earth's intermediate deep subsurface (tens of meters to kilometres), as often done for planning nuclear waste repositories, geothermal projects or tunnelling work, 3D structural modelling commonly starts from a known lithological and structural dataset, may it be from the Earth surface or underground facilities such as tunnels or

boreholes. Known information is then extrapolated towards the unknown. At the time of extrapolation, its validity cannot be proven unless additional information such as geophysical, borehole or excavation data is integrated.

Previous studies report that this extrapolation represents the main uncertainty within 3D structural modelling (e.g. Baumberger, 2015; Bistacchi et al., 2008). More generally, uncertainties in accuracy related to input data (i.e. GPS location, dip / dip azimuth measurements) are small compared to the uncertainty related to the data interpolation between known

locations or to data extrapolation (Bond, 2015).





Uncertainties play an important role when considering decision-making based on information available from a 3D model and have therefore been subject to extensive studies in the past (e.g. Bistacchi et al., 2008; Lindsay et al., 2012; Tacher et al., 2006; Wellmann et al., 2010, 2014; Wellmann and Regenauer-Lieb, 2012; Yamamoto et al., 2014). Since models are a function of the used data, some of the approaches tend to analyse uncertainties of the input data before modelling (e.g. Bond

et al., 2007; Jones et al., 2009). Other approaches investigate the error propagation into the models inferring the uncertainty after modelling (Jessell et al., 2010; Lindsay et al., 2012; Viard et al., 2011; Wellmann et al., 2010). Most of these published studies were performed within sedimentary environments where information such as stratigraphy, layer thickness, layer orientation is known and the overall tectonic setting is rather simple. Uncertainty estimation and its potential reduction are less well constrained for structural modelling of basement rocks (e.g. Svensk Kärnbränslehantering AB, 2009), which are

characterized by intrusive contacts and a complex arrangements of deformation structures.

In this study, we focus on deformed basement rocks and the extrapolation of faults. We follow three main goals: i) Development of an extrapolation workflow for different techniques for projection of surface structures to depth, ii) estimation of related uncertainties using underground information, and iii) design and application of a probabilistic approach to validate the generated model.

We focus specifically on the combination of observations in outcrops at the surface with observations in an underground facility allowing for an extrapolation modelling approach and propose that it is possible to link these two types of observations in a probabilistic context, taking into account uncertainties in measurements, as well as the exact tie between observed features at the surface and in the underground facility. We investigate a local case study in a relatively simple setting in crystalline rocks. The study area is characterized by well-exposed crystalline rocks of the Aar massif in the Central

Swiss Alps (Fig. 1) and furthermore greatly benefits from subsurface information from the Grimsel Test Site (GTS) underground rock laboratory run by the Swiss Cooperative for Disposal of Radioactive Waste (Nagra).

This combination of good outcrop conditions at the surface and independent high quality subsurface information allows for an extrapolation modelling approach and subsequent validation in a relatively simple and well-constrained setting.



## 2. Geological setting

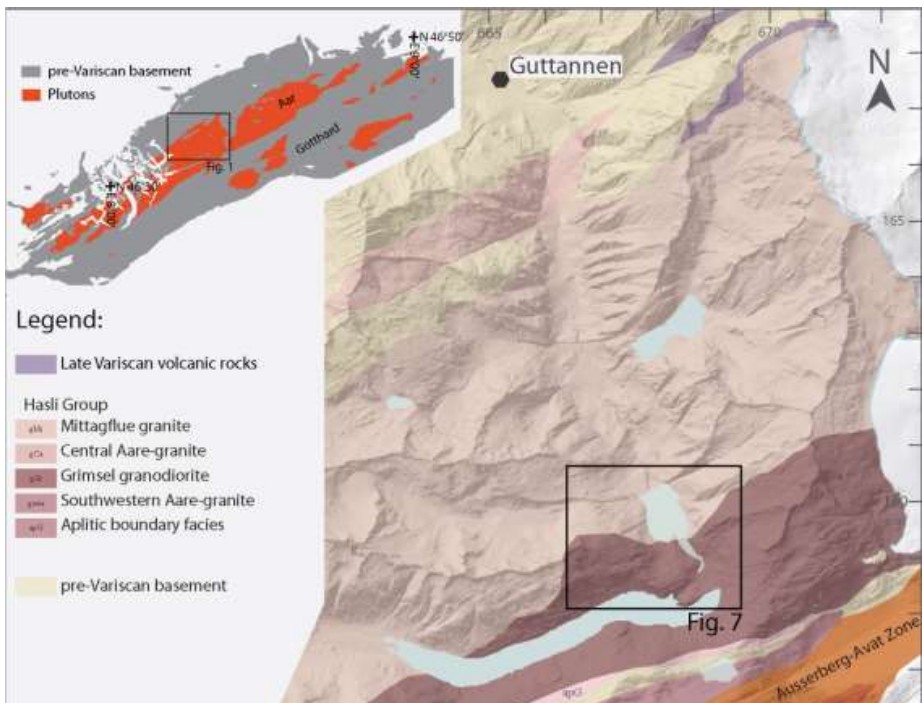

**Figure 1. Geological map of the study area (modified after Berger et al. 2016).**

The study site is located in the Haslital in the Central Alps (Switzerland, Fig. 1) within the Aar massif, an External
Crystalline Massif of the Alps, representing exhumed basement rocks of the former European continental margin and thus
belonging to the paleogeographic Helvetic domain of the Alps (e.g. Mercolli and Oberhänsli, 1988; Pfiffner, 2009; von
Raumer et al., 2009)..

Three different host rocks of magmatic origin occur in the study area: (i) Grimsel granodiorite (GrGr), (ii) Central Aar
granite (CAGr), and (iii) meta-basic dykes (e.g. Abrecht, 1994; Keusen et al., 1989; Labhart, 1977; Stalder, 1964). The GrGr
and the CAGr belong to the Haslital group, which is a Permian calc-alkaline magmatic differentiation suite (Berger et al.,
2016; Schaltegger, 1990; Schaltegger and Corfu, 1992), the GrGr being the more primitive member. The two host rocks
mainly differ in the relative amount of biotite, with ca. 11 vol% biotite in the GrGr compared to ca. 5 vol% biotite in the
CAGr (Keusen et al., 1989). Intermingling structures observed in the field indicate coeval viscous state (Schneeberger et al.,
2016). Furthermore, the concordant zircon and titanite U/Pb intrusion ages of both rock units are overlapping within error,
where the GrGr intrusion has a concordant titanite U/Pb intrusion age of 299±2Ma and the CAGr an age of 299±2Ma
(Schaltegger and Corfu, 1992). The granitoids intruded during Late to Post-Variscan extensional tectonics into a
polymetamorphic pre-Variscan basement (Abrecht, 1994; Berger et al., 2016; Labhart, 1977; von Raumer et al., 2009;
Schaltegger, 1990, 1994).

Meta-basic dykes, formerly called lamprophyres (Oberhänsli, 1986), intrude into the granitoid bedrock without altering the
granitoid indicating only slightly younger intrusion ages of former basic dykes with respect to the calc-alkaline granitoids.

Aforementioned rock types are subsequently overprinted by metamorphism and deformation related to Alpine orogeny. Peak
metamorphic conditions reached 450 ± 30 °C and 6 ± 1 kbar (Challandes et al., 2008) at 22 - 20 Ma (Challandes et al., 2008;
Rolland et al., 2009).





Several authors have described the deformation related to Alpine orogeny in the vicinity of the study area (e.g. Baumberger, 2015; Challandes et al., 2008; Choukroune and Gapais, 1983; Goncalves et al., 2012; Keusen et al., 1989; Marquer et al., 1985; Rolland et al., 2009; Steck, 1968; Wehrens et al., 2016, 2017). Ductile deformation is expressed by a pervasive foliation and by localized high strain zones (shear zones). The exact geometry of the 3D shear zone network, which occurs at

a variety of scales ranging from several kilometres down to millimetres, is complex (Choukroune and Gapais, 1983). It is, however, possible to extract a pattern of km-long major shear zones interconnected by hectometre-long subordinate bridging structures. The major shear zones tend to be quasi-planar (Baumberger, 2015; Wehrens et al., 2017) and we therefore assume a considerably simplified shear zone pattern with quasi-planar to planar geometries of the major shear zones grouped according to their strike orientation (Fig. 2).

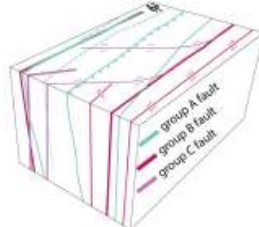

**Figure 2. Schematic bloc diagram showing geometrical relationships between faults of different orientation groups (modified after Wehrens et al. 2017).**

The kinematic framework of Alpine deformation is controversial. Several kinematic models have been proposed for the shear zone network genesis in the study area, including single phase (Choukroune and Gapais, 1983) and multistage

evolution models (Herwegh et al., 2017; Rolland et al., 2009; Steck, 1968; Wehrens et al., 2016, 2017). This study aims at reconstructing the present day 3D geometry and the kinematic evolution is thus of secondary interest. The different orientations of the structures are therefore used without kinematic implications. Major orientation of structures within the area are NE-SW (group A), E-W (group B) and NW-SE (group C) trending (Schneeberger et al., 2016; Wehrens et al., 2017).

The pervasive foliation and the highly localized shear zones form mechanical anisotropies, which favour subsequent brittle localization (Belgrano et al., 2016; Kralik et al., 1992). Deformation in the brittle regime is expressed by fracturing and cataclasis, often resulting in fault gouges (Bense et al., 2014; Wehrens et al., 2016). The spatial distribution of fractures as well as their reactivation in the form of fault gouge development is heterogeneous (Bossart and Mazurek, 1991; Mazurek, 2000).

Although the shear zones experienced a severe ductile deformation history, most of them were reactivated in brittle manner during the exhumation history (Wehrens et al., 2017). Subsequently, we therefore use the term fault as summary terms for high T ductile shear zones, low T ductile shear zones and their reactivation by brittle shearing leading to cohesive (protocataclasite, cataclasite) or non-cohesive (fault gouge) fault rocks.

Present day seismic activity (Pfiffner and Deichmann, 2014) indicate on-going recent tectonic activity in the deep subsurface

of the Aar massif.

Glaciation and glacial retreat contributed to the latest history of the area (Wirsig et al., 2016). Basal erosion and the latest young (17.7 ka, Wirsig et al. 2016) retreat ages produced excellent outcrop conditions, as most outcrops are glacially polished and above the treeline, exposing bare bedrock.

Owing to deglaciation, exfoliation jointing occurred (Ziegler et al., 2013). Given the restricted near-surface occurrence of

these exfoliation joints and their small dimensions, we exclude these deformation features from further consideration in this study.



## 3. Methods

### 3.1. Extrapolation workflow

In order to represent the 3D geometry of faults, we developed a workflow based on a combination of remote sensing and fieldwork (Fig. 3).

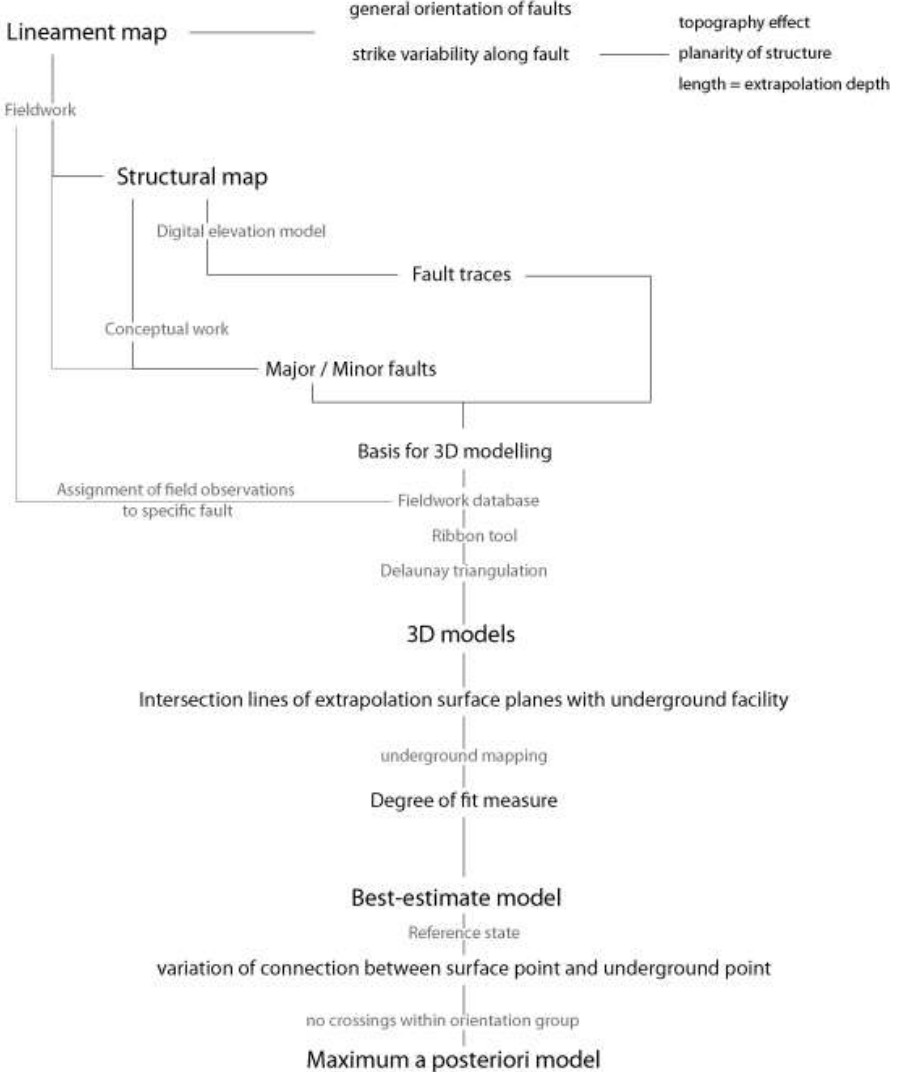

**Figure 3. Modelling workflow**

As a first step, we generated a lineament map using remote sensing data. We use the term lineament as defined by Gabrielsen et al. (2014) and O'Leary et al. (1976), where a lineament is a mappable linear or curvilinear feature identified by remote sensing, possibly representing the intersection between a planar to subplanar structural anisotropy and the Earth

10    surface. Lineament mapping followed the methodology presented by Baumberger (2015). Aerial photographs (Swisstopo) and digital elevation model (DEM; Swisstopo) with resolutions of 0.5 m and 2 m, respectively, served as basis.

Using the DEM, hillshade images (i.e. greyscale relief images) with distinct illumination angles (0°-360° illumination azimuth with 45° steps and constant at 30° altitude angle) were calculated, resulting in eight hillshade images, illuminating different areas of the investigation area. On a pixel-based map, the possible strike angle of a line depends on the number of



pixels of the raster matrix in which the line is enclosed (Heilbronner and Barrett, 2014). Our approach requires an angular resolution <10°, thus a minimum length of 10 pixels for a specific lineament was necessary to fulfil this criterion. Hence, shorter lineaments (< 5 m) were discarded. Lineaments were manually digitized and are composed of minimum 2 endpoints and potentially several points in between.

5 The strike of lineaments was defined as the angle measured clockwise from north. Two different approaches to analyse the strike of lineaments were compared: i) single strike values from endpoint to endpoint, and ii) strike values for individual segments between a lineament's nodes. In both approaches, a weight is added to the strike proportional to the length of the lineament.

In addition to the aforementioned remote sensing approach, conventional structural surface mapping over an area of 13 km$^2$

10 was performed. Spatially restricted outcrop observations at the surface were extrapolated along strike using the lineament map, thus combining fieldwork and remote sensing allowed obtaining a structural surface map. Ductile deformation was mapped differentiating pervasive background strain and localized high strain zones (shear zones). At the surface, mapping of brittle deformation focused on the occurrence of fault gouges. In addition, mapping in the GTS underground facility was performed similarly to surface mapping on dm-scale and in more detail regarding brittle structures (Schneeberger et al.,

15 2016).

Structural modelling was performed using Move$^{TM}$ software (Midland Valley) on two distinct scales: at a local (decametre) scale for the GTS and at a regional scale (km-scale) for the entire study area. Underground 3D structural modelling was performed on the basis of underground mapping and drill core data, which resulted in fault traces and orientations. This information provided the base for the 3D reconstruction of fault planes. Regional 3D structural modelling was performed

20 following published workflows using the surface fault map as a basis (e.g. Baumberger, 2015; Bistacchi et al., 2008; Kaufmann and Martin, 2009; Zanchi et al., 2009). Surface faults were extrapolated to depth by assigning a dip value to individual surface traces, where a trace is the intersection between the Earth's surface and a fault. Three different extrapolation approaches were applied: (i) extrapolation along measured dip and dip azimuth (fieldwork-based approach). Data from outcrops were considered within an orthogonal distance of <20 m to inferred fault traces and a strike differing less

25 than 20° compared to the fault's mean strike as defined by remote sensing. The fault's mean strike was calculated via linear regression through all points defining its trace. (ii) Delaunay triangulation is a 3D meshing algorithm, where triangulation for a given point cloud is calculated such as no point of the point cloud is inside the circumcircle of any triangle connecting three points of the point cloud (Delaunay, 1934). (iii) The ribbon tool is a Move$^{TM}$ internal interpolation algorithm based on a three points approach, where three points form a triangle and the orientation is averaged over a defined number of triangles

30 (Midland Valley). The maximum dip orientation of each average triangle is represented as a stick at the location of the starting point. Combination of all sticks along a trace results in a plane for the given trace. More details on the ribbon tool are given in Fernandez (2005) and Baumberger (2015).

For each approach, the surface fault trace was extrapolated to depth using the obtained specific orientation. Subsequently, the intersection line between the extrapolated plane and a horizontal plane at GTS elevation (approx. 1730 m a.s.l.) was

35 calculated. Then, the resulting intersection lines were compared with the underground structural map in order to find the 'best fitting' underground structure to the obtained intersection line. The degree of fit between the intersection line at the surface and the trace of the underground structure was estimated using the orthogonal distance (distance misfit), starting from the intersection with the main gallery, and the angular difference (angle misfit) between the two linear features (Fig. 4). Only structures within an orientation group (group A, B, C) were compared.



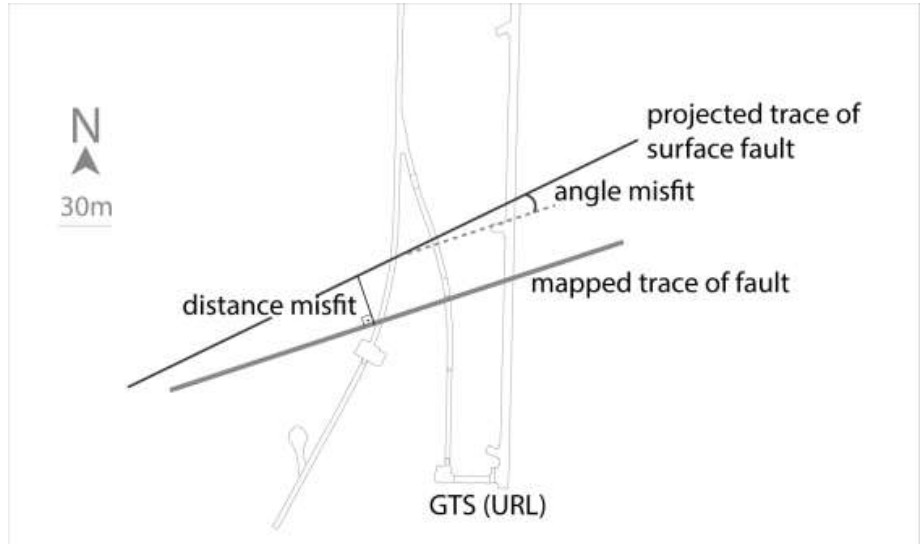

**Figure 4. Schematic drawing of hypothetical example for validation of 3D models based on angular and distance misfit.**

Furthermore, the degree of fit was compared between the different extrapolation approaches and thus for every surface fault. Considering all approaches, a 'best fitting' underground fault was assigned based on the aforementioned criteria. This assignment served as basis for the following structural modelling step, where every surface fault was linearly interpolated with the assigned 'best fitting' underground fault, yielding a 'best estimate' model.

### 3.2. Bayesian Inference

For a better description of the system taking into consideration the inherent uncertainty on the extrapolation methods above, we performed a Bayesian inference. Bayes' theorem

$$p(\theta \,|\, y) = \frac{p(\theta)p(y|\theta)}{p(y)}$$

provides a formal way to update probability distributions for model parameters $\theta$ when new data $y$ is obtained. The final goal is to obtain the posterior distribution $p(\theta \,|\, y)$ of the parameters $\theta$, given the observations y. This distribution is proportional to the distribution of prior parameters $p(\theta)$, and likelihood functions, $p(y|\theta)$ which determine how likely these parameters are, given specific observations $y$. The term $p(y)$ is a normalisation constant and commonly referred to as evidence or marginal likelihood (see for example MacKay, 2003, for more details).

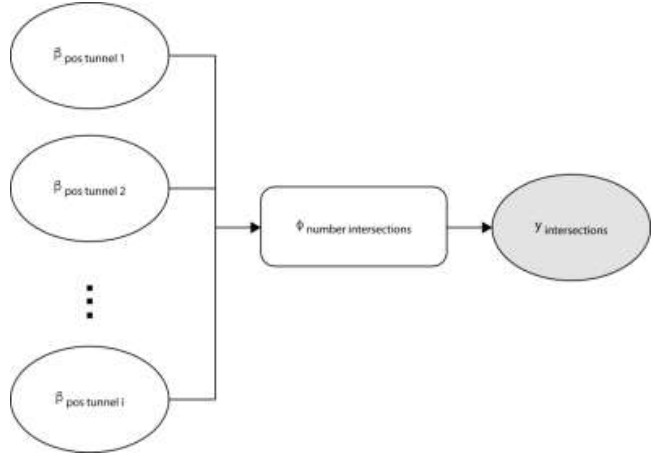

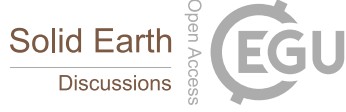

Figure 5. Bayesian inference network scheme

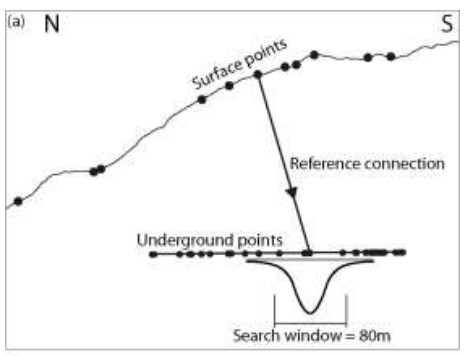

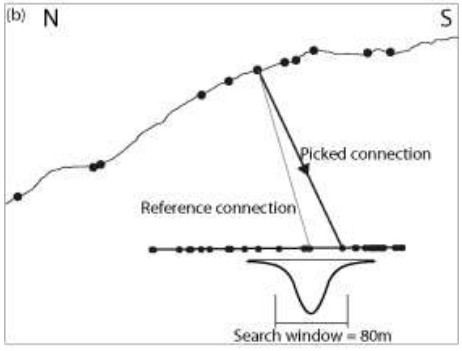

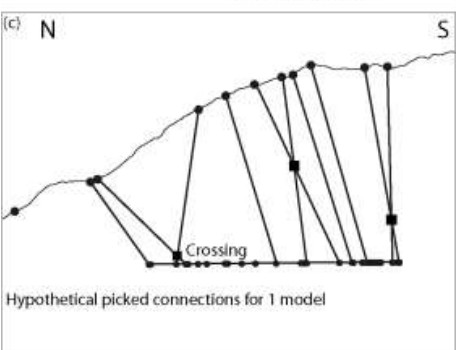

Figure 6. Schematic cross-section illustrating the statistical modelling methodology for one example. (a) Reference state is defined by aforementioned workflow. A search window of 80 m is assigned to the reference underground point. (b) The code picks a possible underground point within the 80 m search window based on a normal Gaussian distribution and calculates the connection. (c) For every surface point a corresponding underground point is picked resulting in a connection pattern. This pattern building is performed 10'000 times and all patterns are compared based on the number of crossings within the specific pattern, yielding in a probability for connecting a certain surface point to a certain underground point

In this study's scenario, we assign a parameter to each surface fault at the tunnel level. We represented the uncertainty about the exact value with a Gaussian distribution and a constant standard deviation of 40 m in the horizontal axis (Fig. 6), based on the dip uncertainty of 10° based on the dip variation in multiple orientation measurements along a single fault (Fig. 6). As a mean value, we assign the 'best estimate' model from the previous interpolation. Interpolated planes were grouped according to their orientation into three separate groups (identified by A, B, C in the following, Fig. 2).

Within each orientation group, we expect faults to be mostly parallel with limited intersections based on field observations. To capture this idea, we assigned a penalty factor that reduces the log-likelihood of a parameter set for an increasing number of intersections (by 0.05 per intersection, to be precise). The number of intersections per iteration was calculated using the Bentley-Ottmann algorithm (Supplement; Balaban, 1995; Bentley and Ottmann, 1979).




The described Bayesian inference cannot be performed directly due to the complexity of multiple parameters in several groups and the non-linearities due to the fault intersections. We therefore apply a computational sampling method based on an Adaptive Metropolis MCMC approach (Haario et al., 2001) implemented in the probabilistic programming package PyMC 2 (Patil et al., 2010).

Final posteriors were discretized to match the locations of measured faults in the underground tunnel by a simple nearest location classifier (Fig. 6). Therefore, the final result of the inference is a discretise distribution of each of the parameters. In order to compare the 3D models obtained by the three extrapolations approaches, we then use the maximum a posteriori value, i.e. the highest probability value of the posterior distributions.

## 4. Results

### 4.1 Lineament map

In total, 5198 lineaments with a spatially heterogeneous distribution and lengths ranging from 5 m to 1941 m were mapped (Fig. 7a). Lineaments are generally more concentrated along topographic highs and lows. Within certain areas (areas (i) and (ii) in Fig. 7a), the lineament's strike tends to be parallel to the dip azimuth of the slopes yielding uniform orientations. In contrast, in domains with relatively low topographic variations a variety of strike orientations become discernible (area (iii)

in Fig. 7a).

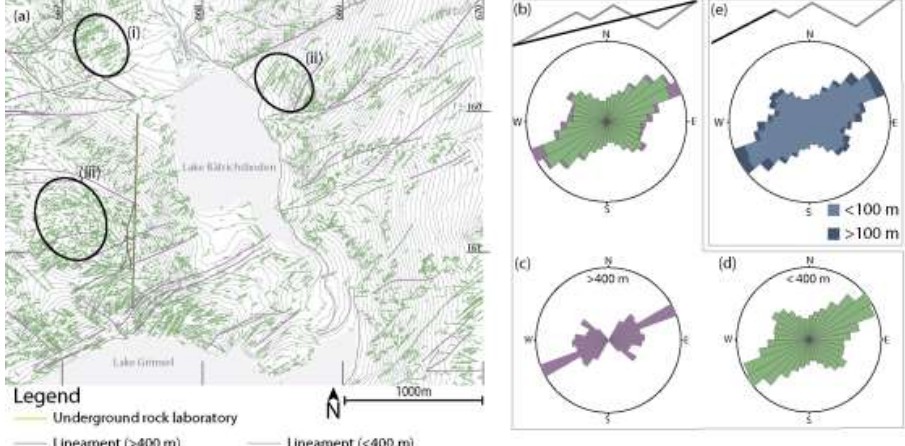

**Figure 7. (a) Lineament map of the study area with underground rock laboratory. Topography contours are based on SwissAlti3D (reproduced by permission of swisstopo (BA17063)). (b-d) Length-weighted rose diagrams showing endpoint to endpoint strike of all lineaments (b), of lineaments longer than 400m (c) and of lineaments shorter than 400m (d). (e) Length-weighted rose diagram**
**showing the orientation of each segment of all lineaments.**

Looking at the bulk data, lineaments show a major NE-SW and a minor NW-SE trend (Fig. 7b). Long lineaments (>400 m) are mainly oriented NE-SW (Fig. 7c), whereas short lineaments show a considerable variation in strike (Fig. 7d). When comparing the two different approaches for strike measurement, the calculated strike distributions are comparable (Figs. 7b and e).

### 4.2 Field observations and data

Data obtained by fieldwork combined with a compilation of several published maps (Baumberger, 2015; Keusen et al., 1989; Vouillomaz, 2009; Wehrens et al., 2017; Wicki, 2011) yielded a surface fault map (Fig. 8, see also Schneeberger et al., 2016).





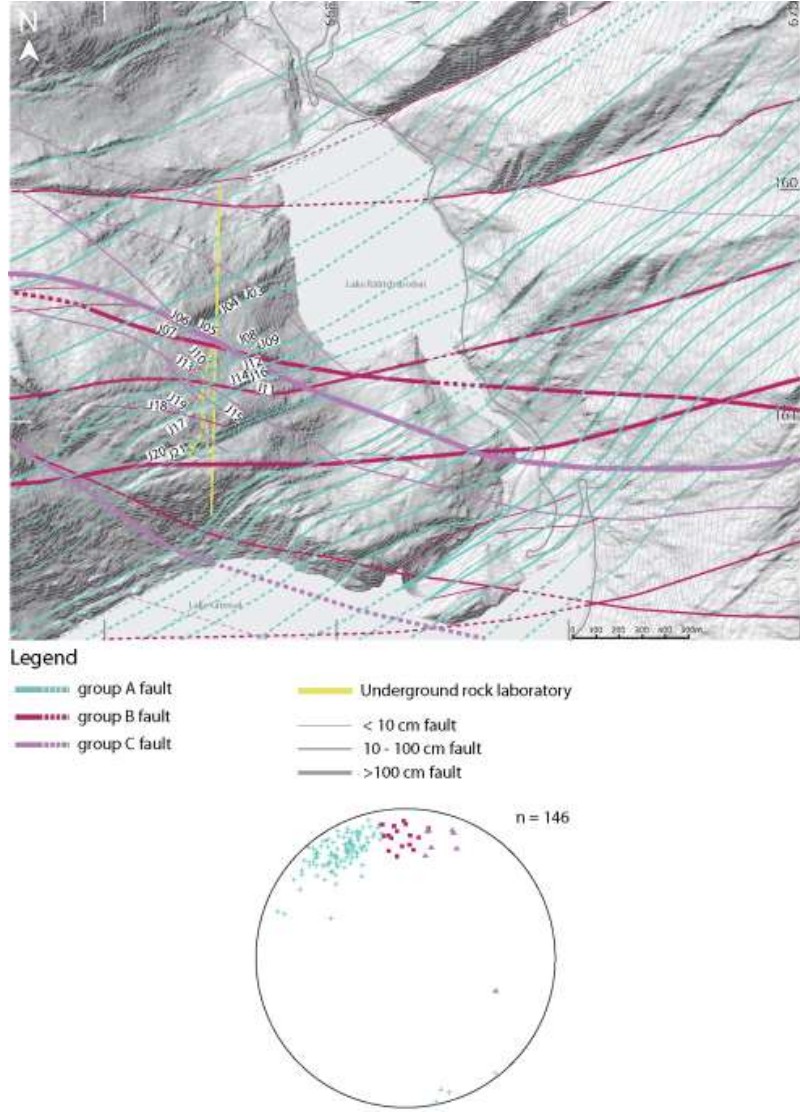

**Figure 8. Surface fault map with faults grouped by strike orientation (group A, B, C). Hillshade image underlying the map is based on SwissAlti3D (reproduced by permission of swisstopo (BA17063)). Fault exposure lines are dashed over uncertain areas and are labelled in cases where a connection to GTS exists. Lower hemisphere equal area projection with planes poles grouped according to strike. Map is based on Swiss coordinate system.**

Based on their orientation, we discriminated different groups of faults (Fig. 8): group A are mainly steep SE-dipping faults. Their average orientation (dip azimuth / dip) is 149/74. Group A faults mostly show steeply plunging stretching-lineations resulting from ductile shearing. Group A can be correlated with faults formed during the Handegg phase (22 – 17 Ma) as defined by Wehrens et al. (2017) while group B and group C would correspond to faults formed during the Oberaar phase (14 – 12 Ma, Wehrens et al., 2017). Group B are mainly steep S-dipping (mean orientation: 178/72) faults. Lastly, group C are SW-dipping faults coeval with group B, with an average orientation of 196/72. Group C faults are subparallel to meta-basic dykes and often co-occur spatially with the latter. Group B and C mostly show oblique to horizontal stretching-lineations. For multiple orientation measurements along individual faults the standard deviation of the mean dip azimuth was below 15° and of the mean dip below 10°. Generally, the GrGr dominated southern area shows an increased number of faults





(Figs 8 and 9). Detailed underground mapping resulted in a lithological (Fig. 9a) and in a structural map of the GTS (Fig. 9b).

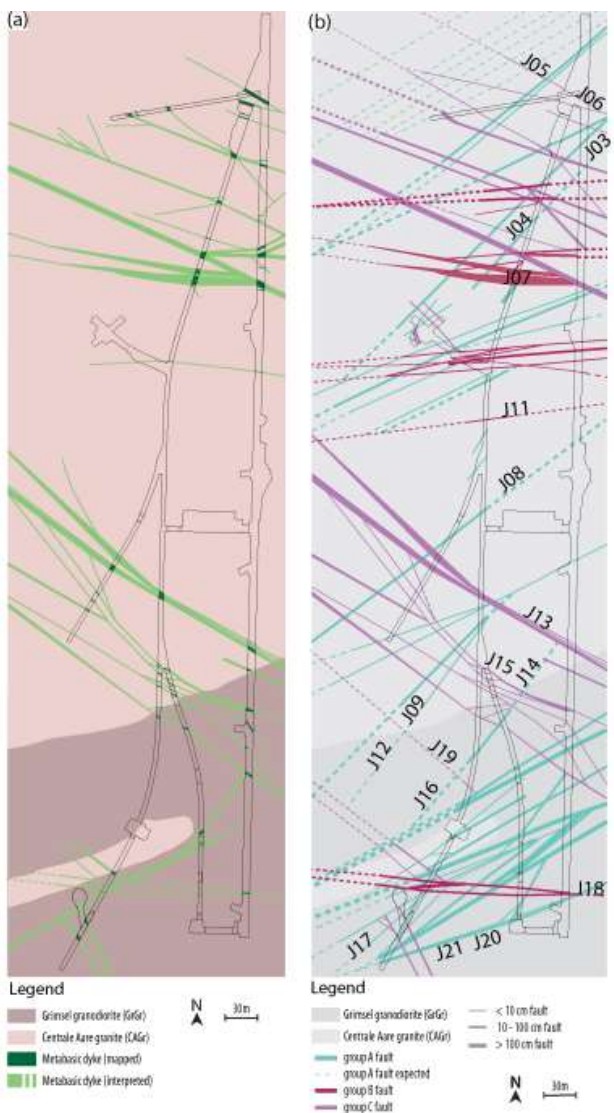

**Figure 9. (a) Petrographic underground map. (b) Structural mapping (1:1000) of the underground rock laboratory (GTS) with faults grouped according to their strike. Indicated labels correspond to surface fault labelling and represent 'maximum a posteriori' interpolation.**

Meta-basic dykes occur as three distinct swarms, two being located within the CAGr domain (Fig. 9a). The northern two swarms strike NW-SE, whereas the southern swarm strikes E-W, however, less clearly marked. Numerous dykes are overprinted by an Alpine foliation, which sometimes is oblique to the dyke boundary. Furthermore, dykes are often overprinted by localized ductile as well as brittle deformation, expressed by shear zones and fault gouges.

Faults occur along three NE-SW trending swarms, two E-W trending swarms and along two NW-SE trending swarms, leading to a heterogeneous strain distribution along the underground facility (Fig. 9b).

The NE-SW trending swarms correspond to group A faults with an average spacing of ca. 16 m. In total 31 group A data were mapped in the underground. They can be further subdivided into 17 moderately to steeply dipping faults (between 45 and 75°) and 14 sub-vertically dipping ones (>80°).



The E-W trending swarms correspond to faults with orientations that are similar to group B. In total, 12 of these E-W striking faults were mapped.

The NW-SE trending fault swarms are localized mainly along dykes (Fig. 9) and represent group C structures. In total, 25 NW-SE striking faults occur within GTS.

Faults in the CAGr (northern part) seem to preferentially localize along pre-existing anisotropies, i.e. high-temperature brittle fractures (biotite coating) or meta-basic dykes and thus form discrete faults (cm-sized) with marked contacts to the host rock. In contrast, faults in the GrGr dominated southern part form strain gradients over larger distances (m-sized). This observation is in agreement with findings of Wehrens et al. (2017).

### 4.3 3D structural modelling

The GTS model size is 600x250x100 m, whereas the regional model size was 4x3 km with a projection depth of 1000 m. The projection depth was defined arbitrarily, but no larger than half of the fault trace's length.

#### 4.3.1 GTS model

Combination of the above-presented underground map with measured surface orientation data resulted in a 3D geometric visualization of meta-basic dykes and faults mapped in the underground. Swarms of meta-basic dykes tend to join towards

less numerous dykes with depth. Based on geometrical considerations, we infer the occurrence of three major dykes from which all others either fan out or form relay structures in-between the major dykes. Based on the field observation that the major faults and relay structures dip steeply to sub-vertically towards the south, we discriminated 8 major group A faults and 23 relay structures. Major group A faults occur within each NE-SW trending swarm discriminated on map view. Group B deformation structures can be further subdivided into 6 major and 7 relay faults. Group C deformation structures can be

subdivided into 6 major and 32 relay deformation structures, some of which are very short (14 m).

#### 4.3.2. Regional model

The surface fault map (Fig. 8) served as a basis for the generation of the three different km-scale 3D models (see above). All three modelling approaches yielded the 3D geometrical visualization of the surface fault pattern. They all share the same fault traces at the model surface. As mentioned above, projection specific dip values were used for each of the models.

However, not all surface faults were extrapolated with each approach. Of the 22 possible surface faults, 15 were extrapolated with the fieldwork-based approach, 12 using the Delaunay triangulation and 21 with the ribbon tool method. Missing projections can be due to lack of outcrop description or absence of sufficient topographic relief for remote sensing based approaches.

By combining all three approaches at least one, but up to three degrees of fit with underground faults were calculated for

each surface fault. Based on the different degrees of fit a 'best fitting' underground structure was assigned to each surface fault. By linearly interpolating the two traces, we obtained a model, which we called "best estimate" model. In total, 11 group A faults reach the GTS. From the total 11, 7 have a dip <80°, which would correspond to the major structures defined in the above presented GTS-scale model, whereas the 4 steeper faults correspond to relay structures. Moreover, 2 group B and 8 group C faults connect the surface with the GTS. Combination of all faults yields an average spacing of 25.4 m.

Furthermore, faults appear to converge with depth.

#### 4.3.3. Bayesian inference

For each model that is obtained when each surface point (intersection between surface fault and 2D section along GTS) is interpolated with a specific underground point, the number of intersections was calculated and the likelihood of the model compiled based on the number of intersections. In total 10'000 models were calculated and for each a probability for a

certain interpolation of a specific surface point with an underground point was obtained (Fig. 10). For certain surface points, a clear maximum a posteriori value was found (Fig.10a-d), however, for other surface points no underground point could be assigned unambiguously (Fig. 10e).





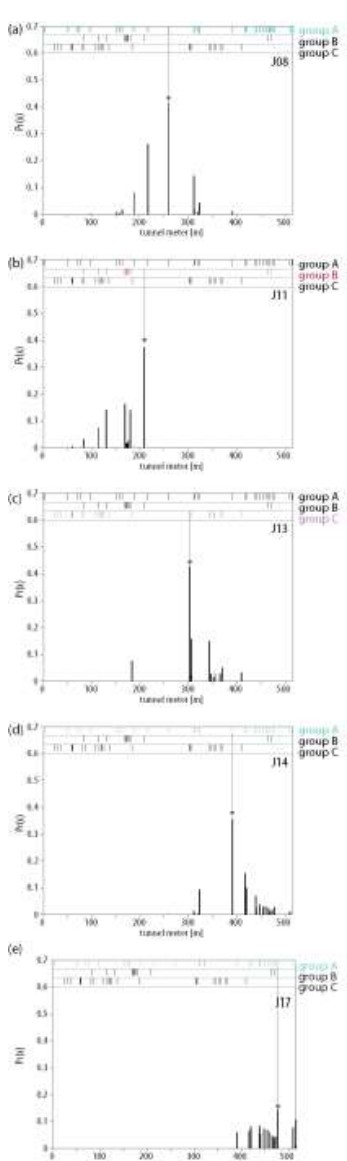

**Figure 10. Probability distributions of five selected examples: (a) to (d) with highest probabilities achieved, whereas (e) shows an example without clear maximum probability. On top are indicated the positions of the underground deformations zones, grouped according to their strike. Additionally, the 'maximum a posteriori' interpolation is highlighted with an arrow.**

5    Based on the maximum a posteriori value a 3D structural model was obtained by linearly interpolating each surface point to the underground point with the maximum a posteriori value. We call this model the 'maximum a posteriori' model (Fig. 11). Notice that the 'maximum a posteriori' model only adds information to the initial model through consideration of a likelihood, i.e. the assumption that crossing faults at large-scale are unlikely. Note that the smaller scaled relay structures are not considered in this approach.



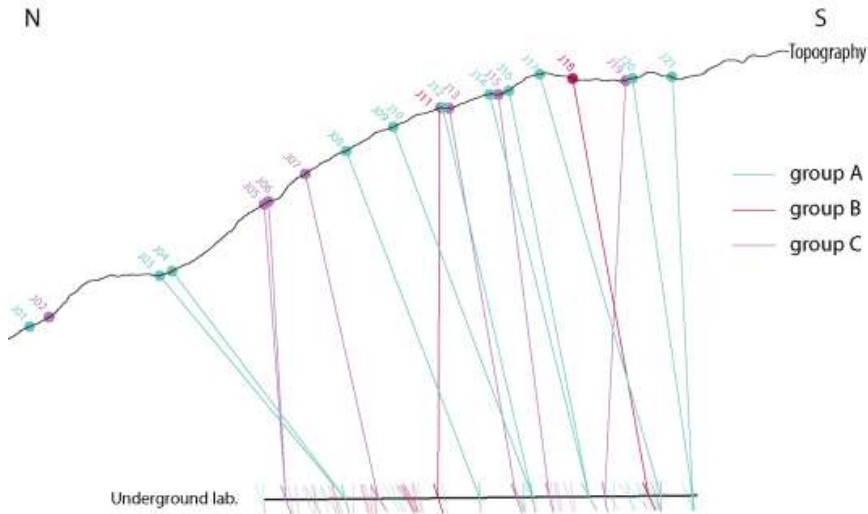

**Figure 11. Cross section showing 'maximum a posteriori' connections between surface and underground faults. Faults are grouped and coloured according to their strike. Underground faults are represented by short ticks where the less transparent are the ones with connection to the surface**

5  This 'maximum a posteriori' interpolation model served as basis for comparing different employed extrapolation approaches. The comparison did not yield a clear 'best' extrapolation approach, however, it seems that fieldwork-based approach results in most accurate extrapolation.

## 5. Discussion

### 5.1 Lineament map

10  Comparison of the remote-sensing based lineament map and field data showed that in intact granitic rocks, purely ductile shear zones without later brittle overprinted are not detected by remote sensing. Brittle deformation generating fractures, cataclasites or even fault gouges responsible for mechanical weakening is necessary to form morphologically detectable structures (Fig. 12a; Baumberger, 2015). Moreover, the orientations of the slopes play an important role, as faults striking down dip directions of slopes are prone to most effective erosion processes driven by gravity. Different orientations observed

15  on the lineament map (Fig. 7a, areas (i) and (ii)) for the eastern and western flank of the Hasli valley are interpreted to result from such preferential erosion. In contrast, surface area (iii) in Fig. 7a is nearly horizontal and thus reflecting a homogeneously eroded pattern of intersection for lineaments. The dependence on erosion for the formation of morphological incisions leads to the observed heterogeneous lineament density distribution as ridges and valleys show higher lineament densities.

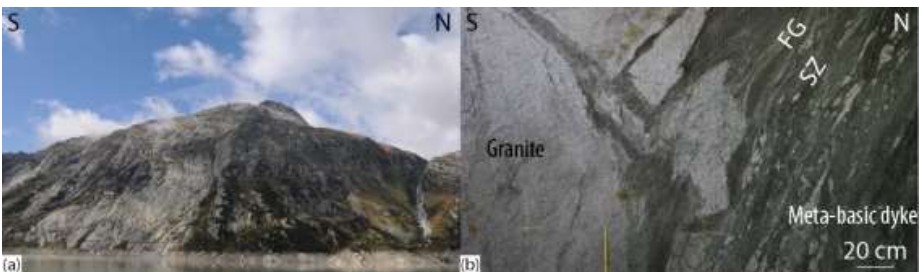



**Figure 12. (a) Mountainside with incisions and exfoliation joints. b) detailed picture of underground outcrop showing outcrop conditions and key structural features:, ductile shear zone (SZ) and a fault gouge (FG).**

Endpoint-to-endpoint strike and strikes of individual segments of lineaments are very similar (Fig. 7b and e) indicating only small variation in strike of the lineaments themselves. Therefore, underlying structures should be quasilinear to linear in 2D

and planar in 3D. We also observe that the longest lineaments are NE-SW striking and that the variability shown in Fig. 7e is mostly due to varying strike orientations of very short lineaments (< 20m). In addition to the NE-SW striking maximum, few long lineaments strike NW-SE. Both major orientations are similar to those reported from field observations (Figs. 8 and 9) and correlate with previous studies (Rolland et al., 2009; Steck, 1968; Wehrens et al., 2017), indicating that lineament maps are suitable to obtain the general trend of steep faults in well exposed crystalline terrain. Much care is needed, however,

when further interpreting lineament maps as the geologic meaning of the lineament is ambiguous and as lineament maps are strongly operator dependent (e.g. Scheiber et al., 2015).

### 5.2. Field observations and data

Differences between the surface map and the underground map are relatively small. The spacing of faults at the surface is lower, but general orientations are comparable (Figs. 8 and 9) and the two mappings are thus discussed conjointly.

Faults commonly show little variation in orientation along strike as evidenced by consistent orientations of dip and dip azimuth of multiple outcrop descriptions along the same fault. In conjunction with the small variability in strike for lineaments, this is clear evidence for the planarity of large-scale faults. At the surface, the 2D length of faults is between 229 and 5591 m (mean 2199 ± 1603 m). Therefore, extrapolation of surface faults to depths similar to depths of the underground faults, which have an overburden between 420 and 520 m, is well in the projection depth range, assuming a circular shape

for the plane as a minimum estimation for their lateral extent.

Localization processes seem to differ between the two host rocks (CAGr and GrGr; Wehrens et al., 2017). The higher amount of biotite in the GrGr could influence the rock's rheology towards more ductile behaviour. In contrast, the relatively higher amount of quartz and K-feldspars renders the CAGr more brittle than GrGr at similar P-T conditions and thus enforces brittle fracturing and possible subsequent ductile shear zone widening, as observed in other crystalline rocks

(Guermani and Pennacchioni, 1998; Mancktelow and Pennacchioni, 2005; Wehrens et al., 2016, 2017). Hence any mechanical anisotropy, such as along pre-existing structures in form of magmatic shear zones, meta-basic dykes or aplitic dykes served in the CAGr as sites for strain localization when suitably oriented with respect to the stress field.

### 5.3. 3D structural modelling

Our 3D structural models were generated as contribution to a project monitoring several parameters such as micro-seismicity

and in-situ stress conditions on the km-scale (Large Scale Monitoring, Nagra). Therefore, 3D structural models were required mostly for visualization purposes. A deterministic explicit modelling workflow was required, as often used in applied projects. It is, however, clear that for model updating an implicit modelling approach would result in faster data handling. The deterministic approach was chosen, because we attempted to obtain a geometrically satisfying product within the simplest geological setting possible, without requiring statistical approximations for representing fault patterns.

Furthermore, we were interested in the actual geometry of the faults dissecting granitoid rock bodies.

#### 5.3.1. Three different approaches to obtain extrapolation 3D structural models (km-scale models)

Uncertainty related to the assignment of specific dip values to lineament traces (Baumberger, 2015; Bistacchi et al., 2008) led to the comparison of three different approaches. Validation attempts by comparison with underground mapping are purely geometrical and were based on two criteria, namely angle and distance misfit (Fig. 4). All three extrapolation

approaches yielded similar results and no significant differences were observed. Moreover, in order to allow for a thorough




comparison between the different extrapolation approaches solely based on the angle and distance misfit, the underground faults would need to be homogeneously distributed, which is not the case (Fig. 9).

The validation procedure could be refined using fault thickness. However, fault thickness varies substantially along strike and thus is not a clear distinction criterion.

In addition to the average dip, the maximum and minimum dips could be used, which would yield a projection cone, similar to uncertainty visualization suggested by Baumberger (2015). Applying this approach to a restricted area such as the underground rock laboratory investigated in this study, resulted in a total coverage and no possible distinction between different faults. However, for a final representation of the uncertainty related to the dip value on a regional scale (km-scale) the approach to visualize projection cones would suit.

**5.3.2. GTS (decametre scale model) compared with km-scale 'best-estimate' model**

As a result of differences in outcrop conditions, the number of observed faults is significantly higher in the underground laboratory compared to the surface (Fig. 13). In the underground, nearly 100% of polished outcrop is accessible along the tunnel walls (Fig. 12b), whereas at the surface faults are often covered with vegetation, even in relatively vegetation-poor domains.

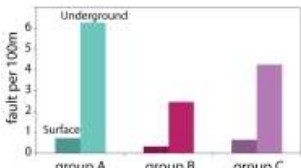

**Figure 13. Histogram showing number of faults grouped per strike at the Earth's surface and in the underground (GTS).**

Furthermore, we observe convergence of surface faults with depth in our 'best estimate' model, which could be a modelling artefact, as only surface faults that occurred spatially close to the underground facility were selected. Subsequently, the degree of fit of each surface fault was calculated to the closest underground fault. Therefore, faults were potentially passively

rotated, yielding a 'pseudo' convergence. Moreover, no similar pattern was observed in the GTS model (decametre scale). We therefore conclude that the 'best estimate' interpolation model suffers from boundary effects, altering the fault orientation at the margin of the GTS and thus only faults in the central part will be further considered.

**5.4.3. 'Maximum a posteriori' model**

Comparison of numerous models obtained from Bayesian inference was performed calculating the number of intersections.

The fewer intersections the more probable the model was considered. Assuming no intersections between large-scale fault set is simplistic, but from field observations seems plausible as a first approach for faults belonging to a specific orientation group (group A, B, C). This simplistic representation of nature enabled us to obtain a probability for all possible interpolations between a specific surface point and all underground points of the corresponding orientation group. As previously mentioned, the margins of the interpolation space show boundary artefacts and thus following surface points at

model margin were not further considered: J03, J04, J05, J06, J19, J20, and J21 (Fig. 9). As expected, probability densities are skewed towards the area of lesser fault density (Fig. 10). At this point it is important to remind that probability density is given as area and therefore we cannot directly compare the discretized posteriors since they are function of the distance between nearby faults.

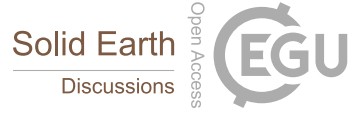

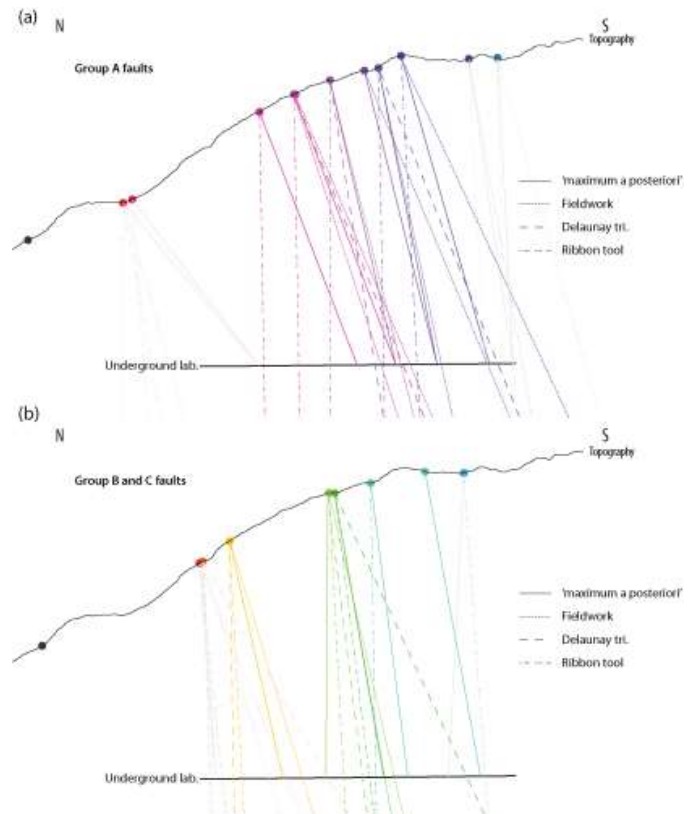

**Figure 14. Comparison of 'maximum a posteriori' interpolation with three extrapolation approaches used to assign dip to fault exposure line. Figure subdivided in: (a) group A (NE-SW) and (b) group B (E-W) and group C (NW-SE). group B and group C are display conjointly as group B contains only two faults.**

5    We compared the initially used three extrapolation techniques based on the 'maximum a posteriori' model (Fig. 14). When comparing group A faults, fieldwork-based extrapolation closely fit the 'maximum a posteriori' interpolation, which indicates either that the fieldwork-based model yields 'best' results or validates the Bayesian inference approach depending if the reference state is the statistical interpolation or the measured field data. Generally, dips of 'maximum a posteriori' models are slightly steeper than measured dips during fieldwork (Fig. 15a). However, the dip differences between the

10   fieldwork-based extrapolation 3D structural model and the 'maximum a posteriori' interpolation model are small (Fig. 15b). Also the dip differences between the ribbon tool based 3D structural model and the 'maximum a posteriori' interpolation model are small, but dips obtained via the ribbon tool are systematically steeper, which do not correspond to the measured dips (Fig. 15a). Extrapolation 3D structural model obtained via Delaunay triangulation is less close to 'maximum a posteriori' interpolation model and obtained dips vary substantially.


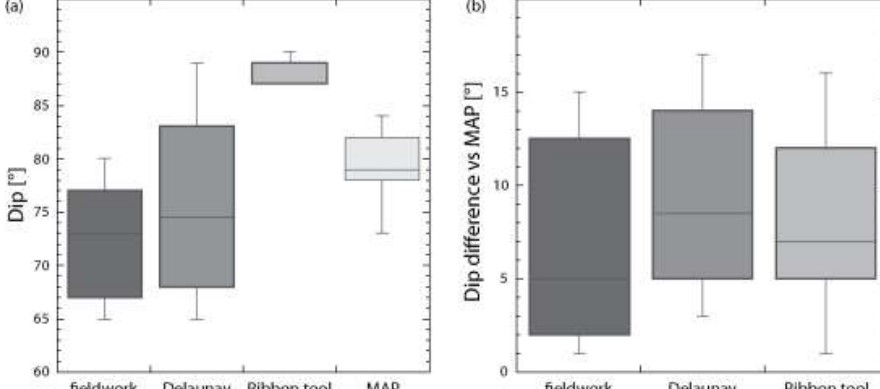

**Figure 15. (a) Box plot showing dip value for different extrapolation approaches and for 'maximum a posteriori' (MAP) interpolation. (b) Box plots for dip comparison between different extrapolation approaches and the 'maximum a posteriori' (MAP) interpolation.**

Comparison for the group B and C faults is less clear. Fieldwork-based and ribbon tool extrapolations are close to the 'maximum a posteriori' model (Fig. 14). Therefore, we conclude that fieldwork is still necessary for 3D structural modelling in crystalline environment and that the ribbon tool (Move$^{TM}$) offers numerous options to tune the obtained plane, however, this tuning requires a profound conceptual background model.

**5.4.4. Possible model refinements**

Presented surface models include only major faults (Fig. 16). However, for further applications such as for example groundwater flow modelling or slip tendency analysis not solely major faults are of interest but also their relay structures. Based on the orientation information gained from the regional km-scale models and on the intersection pattern observed during lineament mapping it is possible to infer a near surface 3D model not only with the major fault but also the relay structure. Furthermore, the increased level of detail in the GTS model (decametre scale) forms a similar model in the

underground. The unknown space between both models would require probabilistic modelling with several key parameters as for example fault spacing, fault orientations, apertures or crosscutting relationships.

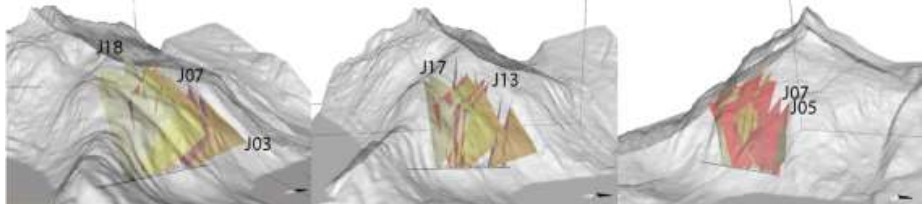

**Figure 16. Representation in 3D of the 'maximum a posteriori' model of fault geometry with three different angle of view. N is indicated by the black triangle. The black tunnel is 717 m long.**

**6. Conclusions**

The exceptional opportunity of surface and underground data comparison over 3D structural modelling approaches led us to the following conclusions:

- Lineament maps enable identification of major fault, but are highly sensitive to preferential erosion.

- Structural surface mapping allowed a discrimination of three orientation groups of faults.

– Comparison based on geometrical criteria (distance and angle misfit) of three approaches to extrapolate to depth surface traces yielded comparable results for all extrapolation approaches and less than 6% distance misfit.



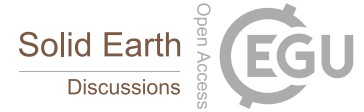

– Interpolation of surface data with underground data based on a Bayesian inference problem showed that the fieldwork-based approach is most accurate extrapolation technique. However, this could also validate the interpolation approach.

We conclude, similarly to Zanchi et al. (2009) that for 3D structural modelling an area within crystalline bedrock classical fieldwork is required not only for field measurements but also as base for a conceptual background model on which interpolations or extrapolations performed within 3D structural modelling can be examined for their validity.

## Competing interest

The authors declare that they have no conflict of interest

## Acknowledgments

This study was funded by the LASMO project run by NAGRA, RWM and SURAO. We thank NAGRA staff in the underground rock laboratory for excellent working environment. We thank the reviewers, (names or anonymous), whose comments greatly improved the manuscript.

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
