# Peer review of "Methods and uncertainty-estimations of 3D structural modelling in crystalline rocks: A case study"

_Solid Earth, 2017_

## Referee Comment (RC1) · C. Bond (Referee) · 5 Jun 2017

Methods and uncertainty-estimations of 3D structural modelling in crystalline rocks: A case study

Schneeberger et al.

General comments The paper is a well-written account of modelling uncertainities in fault continuation into the sub-surface. The authors have the advantage of access to sub-surface tunnels through which they can test their models and assertions. The authors do this using surface data from which they predict and test against the subsurface

tunnel exposure. Due to the nature of the fault systems, actual faults cannot be confidently correlated so the findings are based on probabilistic best-fit models. The result is a deterministic model informed by probability – which is an interesting halfway house between 'normal' deterministic 3D models and probabilistic offerings. The paper topic is suitable for publication in Solid Earth. It reviews well the existing literature in this area of geoscience and introduces the topic well before providing new insight and data to add to geoscience uncertainty literature.

My main thoughts concern the validation process. Given that the spacing of faults in the surface and sub-surface is such that individual faults could not be positively correlated, then the calculated distance misfit of less than 6% for any extrapolation method is partly a function of fault spacing at depth. In reality none of the extrapolations or the associated best-fit models may represent reality. I think this should be considered more explicitly and discussed.

Specific comments

1. Introduction Page 5 line 9 Do you use dip information or purely strike? This makes a big difference in terms of how you can construct a schematic 3d block diagram (Figure 2).

5. Discussion Page 17 lines 19-20 Can you explain/expand the sentence starting "Therefore, faults. . ." So the impact of the modelling and the best-fit model is clear. Where is the centre of the rotation? Is it in a single plane? Do you really mean rotation in this sense?

Page 17 lines 20-22 You introduce the effect of boundary conditions here, and make an assertion in line 20 that because the observations are different in the GTS then there must be boundary condition effects. But in lines 11-14 you discuss differences in the outcrop on the surface and in the GTS, due to vegetation cover, and the ability to identify faults. I think there are modelling reasons why you might have boundary affects. . . you should discuss these, rather than simply saying it doesn't fit out observations so

there must be effects.

Technical corrections

Abstract Page 2 line 4 Insert 'a' between 'as' and 'study area'

1. Introduction Page 3 lines 2-5 The reference to Bond et al. (2007) (as reference in the reference list) should be referenced in line 2. In line 3 the reference should be to Bond et al. (2007b). Knowledge Transfer in a digital world: Field data acquisition, uncertainty and data management. Geosphere 3 (6), 568-575.

2. Geological Setting Page 5 Figure 2 A North arrow would be useful.

Page 5 line 19 I would refer to figure 2 again here.

3. Methods Page 7 line 11 The word 'obtaining ' is in the wring place in the sentence. Change to " . . .remote sensing allowed a structural surface map to be obtained."

Page 7 line 39 Could you expand this sentence to provide a bit more context, again you could refer back to figure 2.

Page 8 Figure 4 Expand the figure caption to discuss the outline of the GTS and the elements annotated in the figure.

Page 9 Figure 5 Expand the figure caption to explain the different elements in the figure.

Page 9 Figure 6 The caption for c) could be clearer, suggest change to ".....10,000 times and all patterns are compared, the number of crossing faults within the generated pattern are minimized by addition a penalty factor yielding a. . ."

Page 9 line 13 The information in brackets is not clear suggest removal of 'in the' to improve clarity.

4. Results Page 12 line 14 Insert "(GTS) facility" at the end of the sentence ". . .mapped in the underground. . .".

Page 13 line 13 Delete "above-presented" it does not add anything and makes the point of the sentence harder to follow.

Page 13 line 14 Ditto remove "mapped in the underground"

5. Discussion Page 18 line 5 This sentence is not well constructed, suggest change to "We initially compared the three extrapolation. . ..."

Page 18 line 13 Add "The" before "extrapolation 3D structural model. . .."

Page 18 line 14 It feels like the end of this sentence needs a bit more. . . "obtained dips vary substantially" from what? How?

---

## Referee Comment (RC2) · Anonymous Referee #2 · 11 Jul 2017

=== Overall quality and general comments ===

I believe that this is a good and well-written work that should be published. An advantage of this paper is the use of high-resolution structural data (within sub-surface tunnel) to validate models built from field data and and to validate modelling techniques. However, there are several corrections that need to be made. Most significantly, a number of the figures need to be improved. Discussion and conclusion should also be rewritten to match the goals presented in the introduction. However, I'm confident that this can be fixed satisfactorily.

=== Specific comments ===

[Figure]

S1: In the introduction, the authors state three main goals (Page 3, line 11): (i) development of an extrapolation workflow for different techniques, (ii) estimation of related uncertainties, (iii) design and application of a probabilistic approach to validate the generated models. The first two goals are not completly clear to me. The third point is clearly tackled within the article and seems to be the most interesting part of the paper. It is not clear to me if the goal is to validate some of the generated models using posterior information (i.e. crossing faults are unlikely), or if the goal is to choose which of the extrapolation technique is the most accurate in this context. I am not sure that the development of extrapolation techniques to build surfaces from points is the goal this paper. Futhermore, the two first points of the conclusion (lineament maps are sensitive to erosion and structural mapping highlight three fault families) are not related to the main subject of the paper. I think refocusing the discussion and the conclusion on the questions stated in the introduction would be beneficial.

S2: The extrapolation methods (Page 7, Line 25) should be described more accurately. Delaunay triangulation is a meshing algorithm that give a triangulation from several points. It provides a surfacic interpolation between points, but no extrapolation. Using only points from the trace of the fault on the surface to compute the dip is highly sensitive to noise (because point are nearly aligned along a line).

S3: Data at depth are not integrated during structural modelling, and are use as posterior data by computing the misfit between observations and model. This is always the case for data that can not be integrated as constraints during structural modelling (flow simulations, geophysical data). However, in this case, fault observations at depth could be integrated as control points during interpolation. The choice to use them as posterior data should be explain.

S4: Figure 2, 5, 7, 9, 10, 11, 14: images and text contain within images are too small to be read. This is a major problem for Figure 10: I can not read the results. 3D illustrations provided in the Supplement are not called within the core of the paper.

= Page 2 = Page 2, line 11: Where does these 6 per cent come from? I only see this number in the abstract and in the conclusion?

Page 2, Line 23: As you use explicit modeling approaches in this article, maybe you could remove the differences between implicit and explicit methods. You could also spend more time reviewing existing stochastic and deterministic modeling methods. This might be usefull before presenting your deterministic modeling method informed by probability.

Page 2, Line 40: "extrapolation represents the main uncertainty within 3D structural modelling". This may not always be the case. Hollund et al, 2002 shows that subseismic faults represent a large uncertainty in flow simulation. More generally, when working with sparse or low-resolution data, the topology of the fault network (number of fault and their connectivity) is highly uncertain and strongly impact flow simulations [Cherpeau et al, 2012; Cherpeau et al, 2015; Julio et al, 2015].

= Page 3 = Line 8: "Most of these published studies were performed where [...] tectonic setting is rather simple". The word "simple" is rather subjective and might offend people who have been/are working on this topic.

Line 12: (TYPO) Development -> development

Line 14: "to validate the generated model". It is not clear to me if you want to validate one of the generated structural models, or if you want to validate one of the 3 structural modeling method (field data/Delaunay/moment of inertia)

= Page 4 = Line 1: title should be after the figure to be visible.

= Page 6 = Figure 3: This figure is not clear. It can not be understood without reading carefully the text. Action verb might could used to distinguish input and output (lineament maps, structural map, best model) from what you do (Modeling 3D structure, Computing degree of misfit...). Caption should be extended.

= Page 7 = Line 3: How many lineaments do you have?

Line 32: (BIBLIO) Fernandez 2005 does not details the Move ribbon tool but a method to find a plane orientation from a point cloud. A distinction between the method, and it software implementation would be more precise. Futhermore, it is always difficult to derive a plane orientation from points aligned along a line (intersection between the topography and the fault). The eigenvalues found should help to detect when point are aligned along a line. This does not seems to be used within the paper (see specific comment S2).

Line 38 and Figure 4: Could you provide the analytical formula used to compute the degree of fit?

= Page 8 = Figure 4: what does URL means?

Figure 4: Could you provide the formula used to derive a degree of fit from angular and distance misfit?

= Page 9 = Figure 6: Is the whole workflow two-dimensional like presented in this Figure, and as suggested by the use of Bentley-Ottman algorithm? In this case, specify it.

= Page 13 =

Line 11: Unclear if the projection depth is defined for the whole model or for each fault. Is it constant for all faults (i.e. =minimum(fault trace's length, 1000m)) or drawn randomly within a distribution?

= Page 14 = Line 8: Why do you make this assumption that faults do not cross at large-scale? Is it common for this geological setting?

= Page 15 = Figure 12: Could you highlight incision and exfoliation joints? Resolution is a low.

= Page 16 = Line 32: Implicit modeling can also be used to extrapolate surface from points, and does not require statistical approximatoin to represents surface. It could

also be used to get the fault geometries. The meaning of these sentences is unclear. See Collon et al., EAGE Paris, 2017 for a short review of Implicit and Explicit modelling (3D Geomodelling in Structurally Complex Areas-Implicit vs. Explicit representations, Collon et al., 2017)

= Page 17 = Line 3: The definitions of fault thickness is ambiguous (does it mean damage zone or fault core as defined by Torabi et al., 2011). Furthermore, if it can not be used in this case, this short paragraph could be removed.

= Page 19 = Figure 15: Acronym MAP is not used, and could be mingled with "map"... Remove it for simplicity.

---

## Referee Comment (RC3) · G. Laurent (Referee) · 13 Jul 2017

**1   General comments**

This paper presents a good application of stochastic modelling of faults to a dataset combining data from surface and subsurface field work. The paper is overall well written and presents an interesting approach to fault modelling. One of the principal assets of this paper is that it combines field work, automated lineament extraction, modelling and a probabilistc study, which has to be acknowledge. The results and conclusion are of interest for the scientific community and mostly well supported by the study. There

is only the conclusion about the misfit being below 6

I am convince that the paper will be of sufficient quality for publication in SE after the comments and corrections have been addressed.

**2  Specific comments**

- Try to go to the point in your introduction. Being too general is always risky as it diverts the read from your main point and takes you into topics that are not directly relevant to your point. Namely here, avoid getting into the explicit/implicit modelling swamp land. I don't agree with the way you explain it and I think one should be very meticulous when tackling this topic becaus eit is quite intricate, and at the same time it is not really the topic of your paper. So I would simply skip it.

- You should reference Cherpeau and Caumon 2015 (10.1144/petgeo2013-030) for the topic of stochastic fault network modelling. It is very similar topic even though the approach is different.

- The quality of pictures is not very good. Try to upload pictures with a higher resolution. Most of them are very difficult to read. It might be due to the sytem and the uploading process. If not, please be careful with te quality of the illustrations.

- page 5, line 15: I disagree with your statement "This study aims at reconstructing the present day 3D geometry and the kinematic evolution is thus of secondary interest." Fault network geometry, topology and kinematics are inter-related and you cannot forget kinematics without willingly missing important information. I understand the kinematics might be quite complex and unresolved in this area, so you might want to consider only current geometry, but you can't suggest that this is just simplifying some details. When you will make decisions about the way faults connect, you will either incorporate kinematics in you reasoning or face the risk to produce models that are kinematically inconsistent.

- fig8: to continue on the remark about kinematics, here you shown different groups of faults, among which B and C, that are presenting horizontal stretching lineations, but there is no trace of any displacement. Do you have an explanation or is it a simplification as well?

- It seems to me that you haven't consider the probability that a fault stops before reaching the GTS or that it is branching on another one. In a sense, you are forcing the between the surface and depth observation and might end up with an overestimation of connectivity. Unless I missed something?

- page 7 line 37: I am not convinced by the distance misfit.this is not well defined... unless your plans are parallels. Are using the center of the segments? But then, is it relevant, because the location of the center would depend primarily on where the tunnels are located?

- Your approach is interesting in the framework of your study, because you have the chance of having both surface and subsurface observations, but your dataset is unique in the sense that we generally don't have such high quality data at depth. Data is generally sparser and less certain. How would your approach be applicable in a more general context? I would suggested to add this discussion to your conclusion.

- Similarly, you have the chance that the faults you are considering seem relatively straight at the scale of your study. How could your approach be applied to more general fault networks?

**3 Technical Corrections**

Further technical corrections are detailed in the attached annotated pdf. Please review them carefully.

Please also note the supplement to this comment:
https://www.solid-earth-discuss.net/se-2017-47/se-2017-47-RC3-supplement.pdf

**Supplement:**

[revised manuscript text omitted]

---

## Author Comment (AC1) · 16 Aug 2017

All comments are numbered in the following manner: X.Xa referee comment X.Xb author response X.Xc changes in the manuscript

The author comments as well as the manuscript with changes highlighted in the tracking mode are provided as supplement.

1. Authors response to interactive comments by Clare Bond on "Methods and uncertainty-estimations of 3D structural modelling in crystalline rocks: A case study" by Raphael Schneeberger et al.

[Figure]

Specific comments

1.1a Introduction Page 5 line 9 Do you use dip information or purely strike? This makes a big difference in terms of how you can construct a schematic 3d block diagram (Figure 2).

1.1b Only the strike of the structures was used to draw the 3d block diagram (Fig. 2). Because the structures are subvertically dipping (see Fig. 7) this simplified approach seems to be justified.

1.2a Discussion Page 17 lines 19-20 Can you explain/expand the sentence starting "Therefore, faults..." So the impact of the modelling and the best-fit model is clear. Where is the centre of the rotation? Is it in a single plane? Do you really mean rotation in this sense?

1.2b We apologize for raising here confusion by using 'rotation'. The term 'rotation' was removed and the sentence rephrased to improve clarity.

1.2c New: The N-S extent of surface area is larger compared to the GTS area, leaving a northern and a southern surface part underneath which no underground data exists (Fig. 10). Hence faults in these domains are forced by the model setup to be connected to the underground, leading to artificial fault orientations in these two cases (N and S rims of Fig. 13a). For that reason, only faults in the central part of the 'best estimate' model will be further considered.

1.3a Page 17 lines 20-22 You introduce the effect of boundary conditions here, and make an assertion in line 20 that because the observations are different in the GTS then there must be boundary condition effects. But in lines 11-14 you discuss differences in the outcrop on the surface and in the GTS, due to vegetation cover, and the ability to identify faults. I think there are modelling reasons why you might have boundary affects. You should discuss these, rather than simply saying it doesn't fit out observations so there must be effects.

1.3b Paragraph was adapted according to the previous comment.

Technical corrections

1.4a Abstract Page 2 line 4 Insert 'a' between 'as' and 'study area'

1.4b 'a' was added as suggested

1.5a Introduction Page 3 lines 2-5 The reference to Bond et al. (2007) (as reference in the reference list) should be referenced in line 2. In line 3 the reference should be to Bond et al. (2007b). Knowledge Transfer in a digital world: Field data acquisition, uncertainty and data management. Geosphere 3 (6), 568-575.

1.5b Reference was changed and added to the reference list

1.6a Geological Setting Page 5 Figure 2 A North arrow would be useful.

1.6b North arrow already existed in the figure but obviously was not visible enough. We enlarged the North arrow.

1.7a Page 5 line 19 I would refer to figure 2 again here.

1.7b Reference was added

1.8a Methods Page 7 line 11 The word 'obtaining ' is in the wring place in the sentence. Change to " ...remote sensing allowed a structural surface map to be obtained."

1.8b Sentence was changed according to suggestion

1.9a Page 7 line 39 Could you expand this sentence to provide a bit more context, again you could refer back to figure 2.

1.9b Sentence was changed in order to clarify the context.

1.9c New: Only structures within the same orientation group (group A, B, C; Fig. 2) were compared.

1.10a Page 8 Figure 4 Expand the figure caption to discuss the outline of the GTS and
the elements annotated in the figure.

1.10b Caption was expanded.

1.10c Schematic drawing of hypothetical example for validation of 3D models based on angular and distance misfit (map view). The contours of the GTS are shown in grey with a mapped fault trace and a fault trace resulting from projection of the fault plane from the surface.

1.11a Page 9 Figure 5 Expand the figure caption to explain the different elements in the figure.

1.11b Based on comment by Gauthier Laurent this figure was deleted.

1.12a Page 9 Figure 6 The caption for c) could be clearer, suggest change to ".....10,000 times and all patterns are compared, the number of crossing faults within the generated pattern are minimized by addition a penalty factor yielding a: : :"

1.12b Caption (c) was reformulated to improve clarity.

1.12c New: (c) For every surface point one underground point is picked and connected by a projection line (hypothetical fault plane). By connecting each surface point with one underground point a connection pattern results. Ten thousand runs were now performed, each connecting different surface and underground points. The 10'000 connection patterns are compared with each other, and evaluated by a penalty criterion. This penalty criterion addresses the number of crossings and rates solutions with lowest number of fault plane crossings highest, yielding in a probability for connecting a specific surface point to a certain underground point.

1.13a Page 9 line 13 The information in brackets is not clear suggest removal of 'in the' to improve clarity.

1.13b Information in brackets was adapted to improve clarity.

1.13c New: (A, B, C; following Fig. 2).

1.14a Results Page 12 line 14 Insert "(GTS) facility" at the end of the sentence "... mapped in the underground...".

1.14b Addition performed based on suggestion

1.15a Page 13 line 13 Delete "above-presented" it does not add anything and makes the point of the sentence harder to follow.

1.15b "above-presented" was removed

1.16a Page 13 line 14 Ditto remove "mapped in the underground"

1.16b "mapped in the underground " was removed.

1.17a Discussion Page 18 line 5 This sentence is not well constructed, suggest change to "We initially compared the three extrapolation: : :.."

1.17b Sentence was reformulated to improve the sentence construction

1.17c New: We compared the three extrapolation techniques based on the 'maximum a posteriori' model (Fig. 14).

1.18a Page 18 line 13 Add "The" before "extrapolation 3D structural model...".

1.18b 'The' was added.

1.19a Page 18 line 14 It feels like the end of this sentence needs a bit more... "obtained dips vary substantially" from what? How?

1.19b End of the sentence was deleted to improve clarity.   2.1 Authors response to interactive comments by an anonymous referee on "Methods and uncertainty-estimations of 3D structural modelling in crystalline rocks: A case study" by Raphael Schneeberger et al.

Specific comments

2.1a S1: In the introduction, the authors state three main goals (Page 3, line 11):

(i) development of an extrapolation workflow for different techniques, (ii) estimation of related uncertainties, (iii) design and application of a probabilistic approach to validate the generated models. The first two goals are not completly clear to me. The third point is clearly tackled within the article and seems to be the most interesting part of the paper. It is not clear to me if the goal is to validate some of the generated models using posterior information (i.e. crossing faults are unlikely), or if the goal is to choose which of the extrapolation technique is the most accurate in this context. I am not sure that the development of extrapolation techniques to build surfaces from points is the goal this paper. Futhermore, the two first points of the conclusion (lineament maps are sensitive to erosion and structural mapping highlight three fault families) are not related to the main subject of the paper. I think refocusing the discussion and the conclusion on the questions stated in the introduction would be beneficial.

2.1b By reading the reviewer's comment we realize that the definition of the primary goal was rather unclear, since it is unclear what different techniques mean, requiring the add 'different techniques for projection of surface structures to depth'. The goals as presented in the introduction were therefore rephrased in order to be more inline with the discussion and the conclusion. Indeed we do not develop extrapolation techniques but we apply already existing techniques and discuss their relevance and quality. It was attempted to compare different extrapolation techniques and possibly gain information from the well-constrained study site for less constrained sites. The two first points of the conclusion are not directly linked to the main focus of the paper, however they are inherently part of a 3D structural modelling exercise, as it would be performed for another study site. Therefore, we think that they should be mentioned within the conclusion but we merged (i) and (ii) since they are ultimately linked to each other.

2.1c New: In this study, we focus on deformed basement rocks and the extrapolation of faults to depth. We follow two main goals: (i) application of an extrapolation workflow for three different techniques for projection of surface structures to depth considering associated projection uncertainties, and (ii) design and application of a probabilistic approach to compare different extrapolation techniques in order to validate the generated models.

2.2a S2: The extrapolation methods (Page 7, Line 25) should be described more accurately. Delaunay triangulation is a meshing algorithm that give a triangulation from several points. It provides a surfacic interpolation between points, but no extrapolation. Using only points from the trace of the fault on the surface to compute the dip is highly sensitive to noise (because point are nearly aligned along a line).

2.2b The description of Delaunay triangulation was expanded as suggested.

2.2c New: (ii) Delaunay triangulation is a meshing algorithm that produces a triangulation for several points such that for a given point cloud no point of the point cloud is inside the circumcircle of any triangle connecting three points of the point cloud (Delaunay, 1934). It results in a 3D surface interpolating the selected points. Based on this 3D surface, the entire fault trace can be extrapolated. Noise can arise because of rugosities of the fault planes, uncertainties of tracing the fault intersection at the surface, and too low vertical variations in topography. In the case of near planar faults, the noise is reduced in the case of high variations in altitude between valleys and mountain peaks as well as by preferring projections of long fault traces over those of short fault segments.

2.3a S3: Data at depth are not integrated during structural modelling, and are use as posterior data by computing the misfit between observations and model. This is always the case for data that can not be integrated as constraints during structural modelling (flow simulations, geophysical data). However, in this case, fault observations at depth could be integrated as control points during interpolation. The choice to use them as posterior data should be explain.

2.3b This is exactly the benefit of the current study. In a majority of cases, the subsurface information is not available. On purpose, we therefore 'blindly' apply the known extrapolation techniques on purpose, without considering the existing depth data, since

we intend to learn more about the real uncertainties related tot the extrapolation. Therefore, we used the underground information solely as posterior in order to closely simulate a common application with the option of validation and assessment for cases without underground information. We added a sentence to the discussion to improve clarity about this point.

2.3c New: Lastly, the uncommonly well-constrained setting of our study site (high-resolution underground data) was used to test and potentially validate extrapolation techniques for common application. Therefore, the underground data was only integrated as validation and not as constrain during interpolation.

2.4a S4: Figure 2, 5, 7, 9, 10, 11, 14: images and text contain within images are too small to be read. This is a major problem for Figure 10: I can not read the results. 3D illustrations provided in the Supplement are not called within the core of the paper.

2.4b Figures adjusted according to graphic requirements of Solid Earth Journal. A sentence was added in chapter 4.3 at page 13 line 11

2.4c New: All 3D models are provided as supplementary material.

Technical corrections

2.5a = Page 2 = Page 2, line 11: Where does these 6 per cent come from? I only see this number in the abstract and in the conclusion?

2.5b The 6% of distance misfit were removed from the manuscript as they were based on the distance misfit, which is methodologically questionable as shown by comments of Clare Bond and Gautier Laurent.

2.6a Page 2, Line 23: As you use explicit modeling approaches in this article, maybe you could remove the differences between implicit and explicit methods. You could also spend more time reviewing existing stochastic and deterministic modeling methods. This might be usefull before presenting your deterministic modeling method informed by probability.

2.6b The introduction paragraph on the differences of explicit and implicit modelling was removed. Further, the paragraph on explicit modelling was extended based on suggested review of existing modelling methods.

2.6c New: Explicit structural modelling can further be subdivided into stochastic and deterministic methods. Deterministic approaches yield a single output for input parameters, analogous to drawing a map, (e.g. Stephens et al., 2015), whereas as in stochastic approaches parameters are defined by a probabilistic density function with a component of randomness (e.g. Cherpeau and Caumon, 2015; González-Garcia and Jessell, 2016; Jørgensen et al., 2015; Koike et al., 2015).

2.7a Page 2, Line 40: "extrapolation represents the main uncertainty within 3D structural modelling". This may not always be the case. Hollund et al, 2002 shows that subseismic faults represent a large uncertainty in flow simulation. More generally, when working with sparse or low-resolution data, the topology of the fault network (number of fault and their connectivity) is highly uncertain and strongly impact flow simulations [Cherpeau et al, 2012; Cherpeau et al, 2015; Julio et al, 2015].

2.7b The sentence was rephrased in order to enhance clarity.

2.7c New: Previous studies report that this extrapolation represents a main uncertainty within 3D structural modelling of known structures (e.g. Baumberger, 2015; Bistacchi et al., 2008). From environments with sparse data, the topology of the fault network is known to be highly uncertain (e.g. Cherpeau et al., 2012; Cherpeau and Caumon, 2015).

2.8a = Page 3 = Line 8: "Most of these published studies were performed where [...] tectonic setting is rather simple". The word "simple" is rather subjective and might offend people who have been/are working on this topic.

2.8b Sentence was rephrased to avoid offending others work.

2.8c New: Most of these published studies were performed within sedimentary environments where parameters such as stratigraphy, layer thickness, layer orientation, and structural setting are well constrained.

2.9a Line 12: (TYPO) Development -> development

2.9b 'Typo' was adapted

2.10a Line 14: "to validate the generated model". It is not clear to me if you want to validate one of the generated structural models, or if you want to validate one of the 3 structural modeling method (field data/Delaunay/moment of inertia)

2.10b Point (iii) of the goals was expanded to clarify the goal

2.10c New: design and application of a probabilistic approach to validate the generated model and to compare different extrapolation techniques

2.11a = Page 4 = Line 1: title should be after the figure to be visible.

2.11b Title was moved.

2.12a = Page 6 = Figure 3: This figure is not clear. It can not be understood without reading carefully the text. Action verb might could used to distinguish input and output (lineament maps, structural map, best model) from what you do (Modeling 3D structure, Computing degree of misfit...). Caption should be extended.

2.12b Figure was adapted and caption extended.

2.12c New: Figure 3. Employed modelling workflow to generate a 3D structural model of the area based on a surface lineament map. As a major step, the workflow also considers the uncertainty related to connection between mapped faults at surface and in the underground.

2.13a = Page 7 = Line 3: How many lineaments do you have?

2.13b 5277 lineaments (Chapter 4.1.)

2.14a Line 32: (BIBLIO) Fernandez 2005 does not details the Move ribbon tool but a

method to find a plane orientation from a point cloud. A distinction between the method, and it software implementation would be more precise. Futhermore, it is always difficult to derive a plane orientation from points aligned along a line (intersection between the topography and the fault). The eigenvalues found should help to detect when point are aligned along a line. This does not seems to be used within the paper (see specific comment S2).

2.14b The sentence was expanded as suggested in order to be more precise. Indeed the eigenvalues are of great help but this is exactly what the Move implementation of Fernandez considers.

2.14c New: More details on the method used by the ribbon tool are given in Fernandez (2005) and Baumberger (2015).

2.15a Line 38 and Figure 4: Could you provide the analytical formula used to compute the degree of fit?

2.15b The degree of fit was measured manually in map view at the level of the underground facility.

2.16a = Page 8 = Figure 4: what does URL means?

2.16b URL was removed from the figure. It stands for underground rock laboratory.

2.17a = Page 9 = Figure 6: Is the whole workflow two-dimensional like presented in this Figure, and as suggested by the use of Bentley-Ottman algorithm? In this case, specify it.

2.17b The workflow for the probabilistic interpolation is two-dimensional.

2.17c New: For a better description of the system taking into consideration the inherent uncertainty on the extrapolation methods above, we performed a Bayesian inference on the basis of a GTS parallel cross-section (Chapter 3.2. - starting sentence was expanded)

2.18a = Page 13 = Line 11: Unclear if the projection depth is defined for the whole model or for each fault. Is it constant for all faults (i.e. =minimum(fault trace's length, 1000m)) or drawn randomly within a distribution?

2.18b The sentence was rephrased to improve clarity.

2.18c New: The GTS model size is 600x250x100 m, whereas the regional model size was 4x3 km with a projection depth reaching the underground facility for all faults.

2.19a = Page 14 = Line 8: Why do you make this assumption that faults do not cross at large-scale? Is it common for this geological setting?

2.19b This simplification was chosen, because observed differences in dip values along large-scale (kilometres) faults were small in our study area. Hence we deal here with large-scale faults (> 800 m) that are subparallel to each other at first glance and are projected over not too long distances (400 to 500 m). This implies that crossings of faults along a cross-section subperpendicular to the strike are improbable. However, it remains a suitable assumption and is kind of a simplification. This is different in the short relay structures (bridges) between the large-scale faults. Here crossings definitely occur. Given the unknown spacing and reoccurrence of these relay structures, we cannot make adequate predictions with the chosen approach, a fact, which we clearly state further down in text.

2.20a = Page 15 = Figure 12: Could you highlight incision and exfoliation joints? Resolution is a low.

2.20b Figure was changed accordingly with addition of some highlighted incisions and exfoliation joints.

2.21a = Page 16 = Line 32: Implicit modeling can also be used to extrapolate surface from points, and does not require statistical approximatoin to represents surface. It could also be used to get the fault geometries. The meaning of these sentences is unclear. See Collon et al., EAGE Paris, 2017 for a short review of Implicit and Explicit modelling (3D Geomodelling in Structurally Complex Areas-Implicit vs. Explicit representations, Collon et al., 2017)

2.21b The end of the sentence was removed as it didn't add information and was not clear.

2.21c New: The deterministic approach was chosen, because we attempted to obtain a geometrically satisfying product within the simplest geological setting possible.

2.22a = Page 17 = Line 3: The definitions of fault thickness is ambiguous (does it mean damage zone or fault core as defined by Torabi et al., 2011). Furthermore, if it can not be used in this case, this short paragraph could be removed.

2.22b We used fault thickness only as the thickness of the fault core.

2.23a = Page 19 = Figure 15: Acronym MAP is not used, and could be mingled with "map"... Remove it for simplicity.

2.23b 'MAP' was removed from the caption in part (b). However, it is kept in part (a) as a box is shown for the dip values obtained for the 'maximum a posteriori' model.

3. Authors response to interactive comments by Gauthier Laurent on "Methods and uncertainty-estimations of 3D structural modelling in crystalline rocks: A case study" by Raphael Schneeberger et al.

Specific comments

3.1a Try to go to the point in your introduction. Being too general is always risky as it diverts the read from your main point and takes you into topics that are not directly relevant to your point. Namely here, avoid getting into the explicit/implicit modelling swamp land. I don't agree with the way you explain it and I think one should be very meticulous when tackling this topic becaus eit is quite intricate, and at the same time it is not really the topic of your paper. So I would simply skip it.

3.1b The introduction was changed based on the reviewer's suggestions

3.2a You should reference Cherpeau and Caumon 2015 (10.1144/petgeo2013-030) for the topic of stochastic fault network modelling. It is very similar topic even though the approach is different.

3.2b Thank you for the suggestion. The reference was added.

3.3a The quality of pictures is not very good. Try to upload pictures with a higher resolution. Most of them are very difficult to read. It might be due to the sytem and the uploading process. If not, please be careful with te quality of the illustrations.

3.3b We checked the images and enhanced the resolution.

3.4a page 5, line 15: I disagree with your statement "This study aims at reconstructing the present day 3D geometry and the kinematic evolution is thus of secondary interest." Fault network geometry, topology and kinematics are inter-related and you cannot forget kinematics without willingly missing important information. I understand the kinematics might be quite complex and unresolved in this area, so you might want to consider only current geometry, but you can't suggest that this is just simplifying some details. When you will make decisions about the way faults connect, you will either incorporate kinematics in you reasoning or face the risk to produce models that are kinematically inconsistent.

3.4b We agree on the reviewer's statement and the importance of kinematics. In our field area, however, the deformation history was rather complex resulting in different episodes of fault reactivation (Baumberger 2015; Wehrens 2015; Wehrens et al. 2016, 2017; Herwegh et al. 2017). Nowadays it is only possible to unravel the general kinematic history of the region but not to resolve the exact kinematics and displacement history of each individual fault. Sentences were rephrased.

3.4c New: This study aims at reconstructing the present day 3D geometry. Although, the kinematic evolution is beyond the scope of this study, the generated models have been validated for kinematic inconsistency with respect to the known tectonic framework (e.g. models with unrealistic dip values have been removed (dip < 60° or north verging)).

3.5a fig8: to continue on the remark about kinematics, here you shown different groups of faults, among which B and C, that are presenting horizontal stretching lineations, but there is no trace of any displacement. Do you have an explanation or is it a simplification as well?

3.5b The apparent absence of displacement is a matter of resolution. Along faults with nearly horizontal stretching lineations displacements were observed, however the offsets were small (decimetres to few meters). Hence these offsets cannot be displayed at the scale on which we were working on. In contrast to individual large-scale faults in other areas (e.g. San Andreas fault system), we are dealing here with a pervasive fault pattern dissecting the entire Aar massif, from which the study area covers only a very small volume). Over the entire massif, each individual fault has accommodated small displacements only but all together accommodated significant vertical displacements of several kilometres. Further elaborations on the kinematic evolution of the study site can be found in Herwegh et al. 2017 and Wehrens et al. 2017.

3.6a It seems to me that you haven't consider the probability that a fault stops before reaching the GTS or that it is branching on another one. In a sense, you are forcing the between the surface and depth observation and might end up with an overestimation of connectivity. Unless I missed something?

3.6b This is an excellent point of the reviewer, which were discussing during the writing stage extensively among us authors. It is true that we forced all surface faults to reach the underground, but please note that only faults longer than 800 m were selected and under the assumption of a circular shape they would reach the underground facility and therefore we think that the simplification that faults go through the entire rock mass to GTS seems feasible. If we would consider greater depths, our assumption would become critical after some depth ranges. Fault continuations at these depth can

only by predicted with some confidence for several kilometres long faults, as done in Baumberger 2015.

3.7a page 7 line 37: I am not convinced by the distance misfit. this is not well defined... unless your plans are parallels. Are using the center of the segments? But then, is it relevant, because the location of the center would depend primarily on where the tunnels are located?

3.7b The distance misfit and the angle misfit are dependent on the orientation of the underground facility and the distance misfit is further a function of the spacing. A homogeneous spacing would be required for a comparison between the measured distance misfits. This is not the case, therefore the angle and the distance misfit were not further considered as validation technique of the 3D model.

3.8a Your approach is interesting in the framework of your study, because you have the chance of having both surface and subsurface observations, but your dataset is unique in the sense that we generally don't have such high quality data at depth. Data is generally sparser and less certain. How would your approach be applicable in a more general context? I would suggested to add this discussion to your conclusion. Similarly, you have the chance that the faults you are considering seem relatively straight at the scale of your study. How could your approach be applied to more general fault networks?

3.8b In terms of general fault networks, our approach can be applied to (i) pervasive regional fault/fracture patterns. Currently it will fail in the case of (ii) discrete large-scale faults (e.g. strike slip faults) consisting of one fault core and associated damage zone. In such cases, more elaborate probabilistic models have to be generated in future, including 3D variations in terms of spacing and orientation of secondary faults and splay faults. For (i) even with few or even missing underground information our approach can be used to predict a surface-based 2D model including a probability evaluation (e.g. variable dip angles) with depth. If available, this evaluation can be tested with individ-

ual depth points such as drill core information. Additionally, an expansion towards 3D would require probability attributes for dip azimuths.

Technical comments

3.9a Page 9, Line 10.

3.9b The constant search window was applied as a first approximation, as the underground facility is nearly horizontal. However, it would be of great interest to apply the Bayesian Inference approach in 3D with the available dataset and use a depth-dependent search window width.

Please also note the supplement to this comment:
https://www.solid-earth-discuss.net/se-2017-47/se-2017-47-AC1-supplement.pdf

**Supplement:**

All comments are numbered in the following manner:

X.Xa referee comment

X.Xb author response

X.Xc changes in the manuscript

1. Authors response to interactive comments by Clare Bond on "Methods and uncertainty-estimations of 3D structural modelling in crystalline rocks: A case study" by Raphael Schneeberger et al.

*Specific comments*

1.1a Introduction Page 5 line 9 Do you use dip information or purely strike? This makes a big difference in terms of how you can construct a schematic 3d block diagram (Figure 2).

1.1b Only the strike of the structures was used to draw the 3d block diagram (Fig. 2). Because the structures are subvertically dipping (see Fig. 7) this simplified approach seems to be justified.

1.2a Discussion Page 17 lines 19-20 Can you explain/expand the sentence starting "Therefore, faults..." So the impact of the modelling and the best-fit model is clear. Where is the centre of the rotation? Is it in a single plane? Do you really mean rotation in this sense?

1.2b We apologize for raising here confusion by using 'rotation'. The term 'rotation' was removed and the sentence rephrased to improve clarity.

1.2c New: The N-S extent of surface area is larger compared to the GTS area, leaving a northern and a southern surface part underneath which no underground data exists (Fig. 10). Hence faults in these domains are forced by the model setup to be connected to the underground, leading to artificial fault orientations in these two cases (N and S rims of Fig. 13a). For that reason, only faults in the central part of the 'best estimate' model will be further considered.

1.3a Page 17 lines 20-22 You introduce the effect of boundary conditions here, and make an assertion in line 20 that because the observations are different in the GTS then there must be boundary condition effects. But in lines 11-14 you discuss differences in the outcrop on the surface and in the GTS, due to vegetation cover, and the ability to identify faults. I think there are modelling reasons why you might have boundary affects. You should discuss these, rather than simply saying it doesn't fit out observations so there must be effects.

1.3b Paragraph was adapted according to the previous comment.

*Technical corrections*

1.4a Abstract Page 2 line 4 Insert 'a' between 'as' and 'study area'

1.4b 'a' was added as suggested

1.5a Introduction Page 3 lines 2-5 The reference to Bond et al. (2007) (as reference in the reference list) should be referenced in line 2. In line 3 the reference should be to Bond et al. (2007b). Knowledge Transfer in a digital world: Field data acquisition, uncertainty and data management. Geosphere 3 (6), 568-575.

1.5b Reference was changed and added to the reference list

1.6a Geological Setting Page 5 Figure 2 A North arrow would be useful.

1.6b North arrow already existed in the figure but obviously was not visible enough. We enlarged the North arrow.

1.7a Page 5 line 19 I would refer to figure 2 again here.

1.7b Reference was added

1.8a Methods Page 7 line 11 The word 'obtaining ' is in the wring place in the sentence. Change to " ...remote sensing allowed a structural surface map to be obtained."

1.8b Sentence was changed according to suggestion

1.9a Page 7 line 39 Could you expand this sentence to provide a bit more context, again you could refer back to figure 2.

1.9b Sentence was changed in order to clarify the context.

1.9c New: Only structures within the same orientation group (group A, B, C; Fig. 2) were compared.

1.10a Page 8 Figure 4 Expand the figure caption to discuss the outline of the GTS and the elements annotated in the figure.

1.10b Caption was expanded.

1.10c Schematic drawing of hypothetical example for validation of 3D models based on angular and distance misfit (map view). The contours of the GTS are shown in grey with a mapped fault trace and a fault trace resulting from projection of the fault plane from the surface.

1.11a Page 9 Figure 5 Expand the figure caption to explain the different elements in the figure.

1.11b Based on comment by Gauthier Laurent this figure was deleted.

1.12a Page 9 Figure 6 The caption for c) could be clearer, suggest change to "....10,000 times and all patterns are compared, the number of crossing faults within the generated pattern are minimized by addition a penalty factor yielding a: : :"

1.12b Caption (c) was reformulated to improve clarity.

1.12c New: (c) For every surface point one underground point is picked and connected by a projection line (hypothetical fault plane). By connecting each surface point with one underground point a connection pattern results. Ten thousand runs were now performed, each connecting different surface and underground points. The 10'000 connection patterns are compared with each other, and evaluated by a penalty criterion. This penalty criterion addresses the number of crossings and rates solutions with lowest number of fault plane crossings highest, yielding in a probability for connecting a specific surface point to a certain underground point.

1.13a Page 9 line 13 The information in brackets is not clear suggest removal of 'in the' to improve clarity.

1.13b Information in brackets was adapted to improve clarity.

1.13c New: (A, B, C; following Fig. 2).

1.14a Results Page 12 line 14 Insert "(GTS) facility" at the end of the sentence "... mapped in the underground...".

1.14b Addition performed based on suggestion

1.15a Page 13 line 13 Delete "above-presented" it does not add anything and makes the point of the sentence harder to follow.

1.15b "above-presented" was removed

1.16a Page 13 line 14 Ditto remove "mapped in the underground"

1.16b "mapped in the underground " was removed.

1.17a Discussion Page 18 line 5 This sentence is not well constructed, suggest change to "We initially compared the three extrapolation: : :.."

1.17b Sentence was reformulated to improve the sentence construction

1.17c New: We compared the three extrapolation techniques based on the 'maximum a posteriori' model (Fig. 14).

1.18a Page 18 line 13 Add "The" before "extrapolation 3D structural model...".

1.18b 'The' was added.

1.19a Page 18 line 14 It feels like the end of this sentence needs a bit more... "obtained dips vary substantially" from what? How?

1.19b End of the sentence was deleted to improve clarity.

2.1 Authors response to interactive comments by an anonymous referee on "Methods and uncertainty-estimations of 3D structural modelling in crystalline rocks: A case study" by Raphael Schneeberger et al.

*Specific comments*

2.1a S1: In the introduction, the authors state three main goals (Page 3, line 11): (i) development of an extrapolation workflow for different techniques, (ii) estimation of related uncertainties, (iii) design and application of a probabilistic approach to validate the generated models. The first two goals are not completly clear to me. The third point is clearly tackled within the article and seems to be the most interesting part of the paper. It is not clear to me if the goal is to validate some of the generated models using posterior information (i.e. crossing faults are unlikely), or if the goal is to choose which of the extrapolation technique is the most accurate in this context. I am not sure that the development of extrapolation techniques to build surfaces from points is the goal this paper. Futhermore, the two first points of the conclusion (lineament maps are sensitive to erosion and structural mapping highlight three fault families) are not related to the main subject of the paper. I think refocusing the discussion and the conclusion on the questions stated in the introduction would be beneficial.

2.1b By reading the reviewer's comment we realize that the definition of the primary goal was rather unclear, since it is unclear what different techniques mean, requiring the add 'different techniques for projection of surface structures to depth'. The goals as presented in the introduction were therefore rephrased in order to be more inline with the discussion and the conclusion. Indeed we do not develop extrapolation techniques but we apply already existing techniques and discuss their relevance and quality. It was attempted to compare different extrapolation techniques and possibly gain information from the well-constrained study site for less constrained sites.
The two first points of the conclusion are not directly linked to the main focus of the paper, however they are inherently part of a 3D structural modelling exercise, as it would be performed for another study site. Therefore, we think that they should be mentioned within the conclusion but we merged (i) and (ii) since they are ultimately linked to each other.

2.1c New: In this study, we focus on deformed basement rocks and the extrapolation of faults to depth. We follow two main goals: (i) application of an extrapolation workflow for three different techniques for projection of surface structures to depth considering associated projection uncertainties, and (ii) design and application of a probabilistic approach to compare different extrapolation techniques in order to validate the generated models.

2.2a S2: The extrapolation methods (Page 7, Line 25) should be described more accurately. Delaunay triangulation is a meshing algorithm that give a triangulation from several points. It provides a surfacic interpolation between points, but no extrapolation. Using only points from the trace of the fault on the surface to compute the dip is highly sensitive to noise (because point are nearly aligned along a line).

2.2b The description of Delaunay triangulation was expanded as suggested.

2.2c New: (ii) Delaunay triangulation is a meshing algorithm that produces a triangulation for several points such that for a given point cloud no point of the point cloud is inside the circumcircle of any triangle connecting three points of the point cloud (Delaunay, 1934). It results in a 3D surface interpolating the selected points. Based on this 3D surface, the entire fault trace can be extrapolated. Noise can arise because of rugosities of the fault planes, uncertainties of tracing the fault intersection at the surface, and too low vertical variations in topography. In the case of near planar faults, the noise is reduced in the case of high variations in altitude between valleys and mountain peaks as well as by preferring projections of long fault traces over those of short fault segments.

2.3a S3: Data at depth are not integrated during structural modelling, and are use as posterior data by computing the misfit between observations and model. This is always the case for data that can not be integrated as constraints during structural modelling (flow simulations, geophysical data). However, in this case, fault observations at depth could be integrated as control points during interpolation. The choice to use them as posterior data should be explain.

2.3b This is exactly the benefit of the current study. In a majority of cases, the subsurface information is

not available. On purpose, we therefore 'blindly' apply the known extrapolation techniques on purpose, without considering the existing depth data, since we intend to learn more about the real uncertainties related tot the extrapolation. Therefore, we used the underground information solely as posterior in order to closely simulate a common application with the option of validation and assessment for cases without underground information.
We added a sentence to the discussion to improve clarity about this point.

2.3c New: Lastly, the uncommonly well-constrained setting of our study site (high-resolution underground data) was used to test and potentially validate extrapolation techniques for common application. Therefore, the underground data was only integrated as validation and not as constrain during interpolation.

2.4a S4: Figure 2, 5, 7, 9, 10, 11, 14: images and text contain within images are too small to be read. This is a major problem for Figure 10: I can not read the results. 3D illustrations provided in the Supplement are not called within the core of the paper.

2.4b Figures adjusted according to graphic requirements of Solid Earth Journal.
A sentence was added in chapter 4.3 at page 13 line 11

2.4c New: All 3D models are provided as supplementary material.

*Technical corrections*

2.5a = Page 2 = Page 2, line 11: Where does these 6 per cent come from? I only see this number in the abstract and in the conclusion?

2.5b The 6% of distance misfit were removed from the manuscript as they were based on the distance misfit, which is methodologically questionable as shown by comments of Clare Bond and Gautier Laurent.

2.6a Page 2, Line 23: As you use explicit modeling approaches in this article, maybe you could remove the differences between implicit and explicit methods. You could also spend more time reviewing existing stochastic and deterministic modeling methods. This might be usefull before presenting your deterministic modeling method informed by probability.

2.6b The introduction paragraph on the differences of explicit and implicit modelling was removed. Further, the paragraph on explicit modelling was extended based on suggested review of existing modelling methods.

2.6c New: Explicit structural modelling can further be subdivided into stochastic and deterministic methods. Deterministic approaches yield a single output for input parameters, analogous to drawing a map, (e.g. Stephens et al., 2015), whereas as in stochastic approaches parameters are defined by a probabilistic density function with a component of randomness (e.g. Cherpeau and Caumon, 2015; González-Garcia and Jessell, 2016; Jørgensen et al., 2015; Koike et al., 2015).

2.7a Page 2, Line 40: "extrapolation represents the main uncertainty within 3D structural modelling". This may not always be the case. Hollund et al, 2002 shows that subseismic faults represent a large uncertainty in flow simulation. More generally, when working with sparse or low-resolution data, the topology of the fault network (number of fault and their connectivity) is highly uncertain and strongly impact flow simulations [Cherpeau et al, 2012; Cherpeau et al, 2015; Julio et al, 2015].

2.7b The sentence was rephrased in order to enhance clarity.

2.7c New: Previous studies report that this extrapolation represents a main uncertainty within 3D structural modelling of known structures (e.g. Baumberger, 2015; Bistacchi et al., 2008). From environments with sparse data, the topology of the fault network is known to be highly uncertain (e.g. Cherpeau et al., 2012; Cherpeau and Caumon, 2015).

2.8a = Page 3 = Line 8: "Most of these published studies were performed where [...] tectonic setting is rather simple". The word "simple" is rather subjective and might offend people who have been/are working on this topic.

2.8b Sentence was rephrased to avoid offending others work.

2.8c New: Most of these published studies were performed within sedimentary environments where parameters such as stratigraphy, layer thickness, layer orientation, and structural setting are well constrained.

2.9a Line 12: (TYPO) Development -> development
2.9b 'Typo' was adapted

2.10a Line 14: "to validate the generated model". It is not clear to me if you want to validate one of the generated structural models, or if you want to validate one of the 3 structural modeling method (field data/Delaunay/moment of inertia)
2.10b Point (iii) of the goals was expanded to clarify the goal
2.10c New: design and application of a probabilistic approach to validate the generated model and to compare different extrapolation techniques

2.11a = Page 4 = Line 1: title should be after the figure to be visible.
2.11b Title was moved.

2.12a = Page 6 = Figure 3: This figure is not clear. It can not be understood without reading carefully the text. Action verb might could used to distinguish input and output (lineament maps, structural map, best model) from what you do (Modeling 3D structure, Computing degree of misfit...). Caption should be extended.
2.12b Figure was adapted and caption extended.
2.12c New: Figure 3. Employed modelling workflow to generate a 3D structural model of the area based on a surface lineament map. As a major step, the workflow also considers the uncertainty related to connection between mapped faults at surface and in the underground.

2.13a = Page 7 = Line 3: How many lineaments do you have?
2.13b 5277 lineaments (Chapter 4.1.)

2.14a Line 32: (BIBLIO) Fernandez 2005 does not details the Move ribbon tool but a method to find a plane orientation from a point cloud. A distinction between the method, and it software implementation would be more precise. Futhermore, it is always difficult to derive a plane orientation from points aligned along a line (intersection between the topography and the fault). The eigenvalues found should help to detect when point are aligned along a line. This does not seems to be used within the paper (see specific comment S2).
2.14b The sentence was expanded as suggested in order to be more precise. Indeed the eigenvalues are of great help but this is exactly what the Move implementation of Fernandez considers.
2.14c New: More details on the method used by the ribbon tool are given in Fernandez (2005) and Baumberger (2015).

2.15a Line 38 and Figure 4: Could you provide the analytical formula used to compute the degree of fit?
2.15b The degree of fit was measured manually in map view at the level of the underground facility.

2.16a = Page 8 = Figure 4: what does URL means?
2.16b URL was removed from the figure. It stands for underground rock laboratory.

2.17a = Page 9 = Figure 6: Is the whole workflow two-dimensional like presented in this Figure, and as suggested by the use of Bentley-Ottman algorithm? In this case, specify it.
2.17b The workflow for the probabilistic interpolation is two-dimensional.
2.17c New: For a better description of the system taking into consideration the inherent uncertainty on the extrapolation methods above, we performed a Bayesian inference on the basis of a GTS parallel cross-section (Chapter 3.2. - starting sentence was expanded)

2.18a = Page 13 = Line 11: Unclear if the projection depth is defined for the whole model or for each fault. Is it constant for all faults (i.e. =minimum(fault trace's length, 1000m)) or drawn randomly within a distribution?
2.18b The sentence was rephrased to improve clarity.
2.18c New: The GTS model size is 600x250x100 m, whereas the regional model size was 4x3 km with

a projection depth reaching the underground facility for all faults.

2.19a = Page 14 = Line 8: Why do you make this assumption that faults do not cross at large-scale? Is it common for this geological setting?

2.19b This simplification was chosen, because observed differences in dip values along large-scale (kilometres) faults were small in our study area. Hence we deal here with large-scale faults (> 800 m) that are subparallel to each other at first glance and are projected over not too long distances (400 to 500 m). This implies that crossings of faults along a cross-section subperpendicular to the strike are improbable. However, it remains a suitable assumption and is kind of a simplification. This is different in the short relay structures (bridges) between the large-scale faults. Here crossings definitely occur. Given the unknown spacing and reoccurrence of these relay structures, we cannot make adequate predictions with the chosen approach, a fact, which we clearly state further down in text.

2.20a = Page 15 = Figure 12: Could you highlight incision and exfoliation joints? Resolution is a low.

2.20b Figure was changed accordingly with addition of some highlighted incisions and exfoliation joints.

2.21a = Page 16 = Line 32: Implicit modeling can also be used to extrapolate surface from points, and does not require statistical approximatoin to represents surface. It could also be used to get the fault geometries. The meaning of these sentences is unclear. See Collon et al., EAGE Paris, 2017 for a short review of Implicit and Explicit modelling (3D Geomodelling in Structurally Complex Areas-Implicit vs. Explicit representations, Collon et al., 2017)

2.21b The end of the sentence was removed as it didn't add information and was not clear.

2.21c New: The deterministic approach was chosen, because we attempted to obtain a geometrically satisfying product within the simplest geological setting possible.

2.22a = Page 17 = Line 3: The definitions of fault thickness is ambiguous (does it mean damage zone or fault core as defined by Torabi et al., 2011). Furthermore, if it can not be used in this case, this short paragraph could be removed.

2.22b We used fault thickness only as the thickness of the fault core.

2.23a = Page 19 = Figure 15: Acronym MAP is not used, and could be mingled with "map"... Remove it for simplicity.

2.23b 'MAP' was removed from the caption in part (b). However, it is kept in part (a) as a box is shown for the dip values obtained for the 'maximum a posteriori' model.

3. Authors response to interactive comments by Gauthier Laurent on "Methods and uncertainty-estimations of 3D structural modelling in crystalline rocks: A case study" by Raphael Schneeberger et al.

*Specific comments*

3.1a Try to go to the point in your introduction. Being too general is always risky as it diverts the read from your main point and takes you into topics that are not directly relevant to your point. Namely here, avoid getting into the explicit/implicit modelling swamp land. I don't agree with the way you explain it and I think one should be very meticulous when tackling this topic becaus eit is quite intricate, and at the same time it is not really the topic of your paper. So I would simply skip it.

3.1b The introduction was changed based on the reviewer's suggestions

3.2a You should reference Cherpeau and Caumon 2015 (10.1144/petgeo2013-030) for the topic of stochastic fault network modelling. It is very similar topic even though the approach is different.

3.2b Thank you for the suggestion. The reference was added.

3.3a The quality of pictures is not very good. Try to upload pictures with a higher resolution. Most of them are very difficult to read. It might be due to the sytem and the uploading process. If not, please be careful with te quality of the illustrations.

3.3bWe checked the images and enhanced the resolution.

3.4a page 5, line 15: I disagree with your statement "This study aims at reconstructing the present day 3D geometry and the kinematic evolution is thus of secondary interest." Fault network geometry, topology and kinematics are inter-related and you cannot forget kinematics without willingly missing important information. I understand the kinematics might be quite complex and unresolved in this area, so you might want to consider only current geometry, but you can't suggest that this is just simplifying some details. When you will make decisions about the way faults connect, you will either incorporate kinematics in you reasoning or face the risk to produce models that are kinematically inconsistent.

3.4b We agree on the reviewer's statement and the importance of kinematics. In our field area, however, the deformation history was rather complex resulting in different episodes of fault reactivation (Baumberger 2015; Wehrens 2015; Wehrens et al. 2016, 2017; Herwegh et al. 2017). Nowadays it is only possible to unravel the general kinematic history of the region but not to resolve the exact kinematics and displacement history of each individual fault. Sentences were rephrased.

3.4c New: This study aims at reconstructing the present day 3D geometry. Although, the kinematic evolution is beyond the scope of this study, the generated models have been validated for kinematic inconsistency with respect to the known tectonic framework (e.g. models with unrealistic dip values have been removed (dip < 60° or north verging)).

3.5a fig8: to continue on the remark about kinematics, here you shown different groups of faults, among which B and C, that are presenting horizontal stretching lineations, but there is no trace of any displacement. Do you have an explanation or is it a simplification as well?

3.5b The apparent absence of displacement is a matter of resolution. Along faults with nearly horizontal stretching lineations displacements were observed, however the offsets were small (decimetres to few meters). Hence these offsets cannot be displayed at the scale on which we were working on. In contrast to individual large-scale faults in other areas (e.g. San Andreas fault system), we are dealing here with a pervasive fault pattern dissecting the entire Aar massif, from which the study area covers only a very small volume). Over the entire massif, each individual fault has accommodated small displacements only but all together accommodated significant vertical displacements of several kilometres. Further elaborations on the kinematic evolution of the study site can be found in Herwegh et al. 2017 and Wehrens et al. 2017.

3.6a It seems to me that you haven't consider the probability that a fault stops before reaching the GTS or that it is branching on another one. In a sense, you are forcing the between the surface and depth observation and might end up with an overestimation of connectivity. Unless I missed something?

3.6b This is an excellent point of the reviewer, which were discussing during the writing stage

extensively among us authors. It is true that we forced all surface faults to reach the underground, but please note that only faults longer than 800 m were selected and under the assumption of a circular shape they would reach the underground facility and therefore we think that the simplification that faults go through the entire rock mass to GTS seems feasible. If we would consider greater depths, our assumption would become critical after some depth ranges. Fault continuations at these depth can only by predicted with some confidence for several kilometres long faults, as done in Baumberger 2015.

3.7a page 7 line 37: I am not convinced by the distance misfit. this is not well defined... unless your plans are parallels. Are using the center of the segments? But then, is it relevant, because the location of the center would depend primarily on where the tunnels are located?

3.7b The distance misfit and the angle misfit are dependent on the orientation of the underground facility and the distance misfit is further a function of the spacing. A homogeneous spacing would be required for a comparison between the measured distance misfits. This is not the case, therefore the angle and the distance misfit were not further considered as validation technique of the 3D model.

3.8a Your approach is interesting in the framework of your study, because you have the chance of having both surface and subsurface observations, but your dataset is unique in the sense that we generally don't have such high quality data at depth. Data is generally sparser and less certain. How would your approach be applicable in a more general context? I would suggested to add this discussion to your conclusion.

Similarly, you have the chance that the faults you are considering seem relatively straight at the scale of your study. How could your approach be applied to more general fault networks?

3.8b In terms of general fault networks, our approach can be applied to (i) pervasive regional fault/fracture patterns. Currently it will fail in the case of (ii) discrete large-scale faults (e.g. strike slip faults) consisting of one fault core and associated damage zone. In such cases, more elaborate probabilistic models have to be generated in future, including 3D variations in terms of spacing and orientation of secondary faults and splay faults. For (i) even with few or even missing underground information our approach can be used to predict a surface-based 2D model including a probability evaluation (e.g. variable dip angles) with depth. If available, this evaluation can be tested with individual depth points such as drill core information. Additionally, an expansion towards 3D would require probability attributes for dip azimuths.

*Technical comments*

3.9a Page 9, Line 10.

3.9b The constant search window was applied as a first approximation, as the underground facility is nearly horizontal. However, it would be of great interest to apply the Bayesian Inference approach in 3D with the available dataset and use a depth-dependent search window width.

[revised manuscript text omitted]

5  **Figure 4. Schematic drawing of hypothetical example for validation of 3D models based on angular and distance misfit (map view). The contours of the GTS are shown in grey with a mapped fault trace and a fault trace resulting from projection of the fault plane from the surface.**

Furthermore, the degree of fit was compared between the different extrapolation approaches and thus for every surface fault. Considering all approaches, a 'best fitting' underground fault was assigned based on the aforementioned criteria. This
10  assignment served as basis for the following structural modelling step, where every surface fault was linearly interpolated with the assigned 'best fitting' underground fault, yielding a 'best estimate' model.

**3.2. Bayesian Inference**

For a better description of the system taking into consideration the inherent uncertainty on the extrapolation methods above, we performed a Bayesian inference on the basis of a GTS parallel cross-section. Bayes' theorem

$$p(\theta \mid y) = \frac{p(\theta)p(y|\theta)}{p(y)}$$

15  provides a formal way to update probability distributions for model parameters $\theta$ when new data $y$ is obtained. The final goal is to obtain the posterior distribution $p(\theta \mid y)$ of the parameters $\theta$, given the observations y. This distribution is proportional to the distribution of prior parameters $p(\theta)$, and likelihood functions, $p(y|\theta)$ which determine how likely these parameters are, given specific observations $y$. The term $p(y)$ is a normalisation constant and commonly referred to as evidence or marginal likelihood (see for example MacKay, 2003, for more details).

[Figure]

Raphael Schneeberger 4.8.2017 09:05

Raphael Schneeberger 4.8.2017 09:45

Raphael Schneeberger 14.8.2017 08:32

Raphael Schneeberger 4.8.2017 09:05

[Figure]

Figure 5. Schematic cross-section illustrating the statistical modelling methodology for one example. (a) Reference state is defined by aforementioned workflow. A search window of 80 m is assigned to the reference underground point. (b) The code picks a possible underground point within the 80 m search window based on a normal Gaussian distribution and calculates the connection. (c) For every surface point one underground point is picked and connected by a projection line (hypothetical fault plane). By connecting each surface point with one underground point a connection pattern results. Ten thousand runs were now performed, each connecting different surface and underground points. The 10'000 connection patterns are compared with each other, and evaluated by a penalty criterion. This penalty criterion addresses the number of crossings and rates solutions with lowest number of fault plane crossings highest, yielding in a probability for connecting a specific surface point to a certain underground point.

[revised manuscript text omitted]

25 Furthermore, we observe convergence of surface faults with depth in our 'best estimate' model, which could be a modelling artefact. The N-S extent of surface area is larger compared to the GTS area, leaving a northern and a southern surface part underneath which no underground data exists (Fig. 10). Hence faults in these domains are forced by the model setup to be connected to the underground, leading to artificial fault orientations in these two cases (N and S rims of Fig. 13a). For that reason, only faults in the central part of the 'best estimate' model will be further considered.

30 **5.4.3. 'Maximum a posteriori' model**

Comparison of numerous models obtained from Bayesian inference was performed calculating the number of intersections. The fewer intersections the more probable the model was considered. Assuming no intersections within large-scale fault set is simplistic, but from field observations seems plausible (Fig. 7) as a first approach for faults belonging to a specific orientation group (group A, B, C). The 'maximum a posteriori' model is based on a N-S cross-section along the GTS, this

35 orientation implies that fault would only cross if their dip varies strongly. Such a strong variation in the dip value is improbable based on measured dip values (Fig. 7). Therefore, this assumption seems feasible.

Raphael Schneeberger 4.8.2017 09:09

Raphael Schneeberger 4.8.2017 09:09

Raphael Schneeberger 4.8.2017 09:09

Raphael Schneeberger 4.8.2017 10:01

[Figure]

Raphael Schneeberger 4.8.2017 09:09

Raphael Schneeberger 4.8.2017 09:17

Raphael Schneeberger 4.8.2017 09:17

[revised manuscript text omitted]

We conclude, similarly to Zanchi et al. (2009) that for 3D structural modelling a high-topography area within crystalline bedrock classical fieldwork as information source and as base for a conceptual background model on which interpolations or extrapolations performed within 3D structural modelling can be examined for their validity. In terms of general fault networks, our approach can be applied to (i) pervasive regional fault/fracture patterns. Currently it will fail in the case of (ii) discrete large-scale faults (e.g. strike slip faults) consisting of one fault core and associated damage zone. In such cases, more elaborate probabilistic models have to be generated in future, including 3D variations in terms of spacing and orientation of secondary faults and splay faults. For (i) even with few or even missing underground information our approach can be used to predict a surface-based 2D model including a probability evaluation (e.g. variable dip angles) with depth. If available, this evaluation can be tested with individual depth points such as drill core information. Additionally, an expansion towards 3D would require probability attributes for dip azimuths.

**Competing interest**

The authors declare that they have no conflict of interest

**Acknowledgments**

This study was funded by the LASMO project run by NAGRA, RWM and SURAO. We thank NAGRA staff in the underground rock laboratory for excellent working environment. We thank the reviewers, Clare Bond, Gautier Laurent and an anonymous reviewer, whose comments greatly improved the manuscript.